# Chaos Meets Attention: Transformers for Large-Scale Dynamical Prediction

Yi He [* 1]   Yiming Yang [* 1 2]   Xiaoyuan Cheng [* 1]   Hai Wang [2]   Xiao Xue [3]   Boli Chen [4]   Yukun Hu [1]

## Abstract

Generating long-term trajectories of dissipative chaotic systems autoregressively is a highly challenging task. The inherent positive Lyapunov exponents amplify prediction errors over time. Many chaotic systems possess a crucial property — ergodicity on their attractors, which makes long-term prediction possible. State-of-the-art methods address ergodicity by preserving statistical properties using optimal transport techniques. However, these methods face scalability challenges due to the curse of dimensionality when matching distributions. To overcome this bottleneck, we propose a scalable transformer-based framework capable of stably generating long-term high-dimensional and high-resolution chaotic dynamics while preserving ergodicity. Our method is grounded in a physical perspective, revisiting the Von Neumann mean ergodic theorem to ensure the preservation of long-term statistics in the $\mathcal{L}^2$ space. We introduce novel modifications to the attention mechanism, making the transformer architecture well-suited for learning large-scale chaotic systems. Compared to operator-based and transformer-based methods, our model achieves better performances across five metrics, from short-term prediction accuracy to long-term statistics. In addition to our methodological contributions, we introduce new chaotic system benchmarks: a machine learning dataset of $140k$ snapshots of turbulent channel flow and a processed high-dimensional Kolmogorov Flow dataset, along with various evaluation metrics for both short- and long-term performances. Both are well-suited for machine learning research on chaotic systems.

*Equal contribution [1]Dynamic Systems, University College London, United Kingdom [2]Statistical Science, University College London, United Kingdom [3]Center for Computational Science, University College London, United Kingdom [4]Electronic and Electrical Engineering, University College London, United Kingdom. Correspondence to: Yukun Hu <yukun.hu@ucl.ac.uk>.

*Proceedings of the $42^{nd}$ International Conference on Machine Learning*, Vancouver, Canada. PMLR 267, 2025. Copyright 2025 by the author(s).

## 1. Introduction

Predicting and modeling the behavior of chaotic systems is crucial for various applications, including weather forecasting (Lorenz & Haman, 1996), climate modeling (Trevisan & Palatella, 2011), and understanding turbulent flows (Ottino et al., 1990). In recent years, the use of autoregressive models to predict the long-term behavior of chaotic systems has presented a significant challenge to the ML/DL community (Gilpin, 2021; Mikhaeil et al., 2022). This challenge stems from the presence of positive Lyapunov exponents, a hallmark of chaotic systems. These exponents amplify small perturbations in the initial state, leading to exponential accumulation of errors over time (Holden, 2014). Consequently, accurately predicting the long-term behavior of chaotic systems remains a formidable task. To address these challenges, two primary methods and one emerging trend have been reviewed for learning chaotic systems: a) sequence models that excel in learning temporal patterns; b) operator-based models, which learn operators in function spaces without knowing the prior differential equations; and c) the transformer-based models, with recent advancements in synthesizing sequence and operator-based methods.

**Sequence models.** Sequence models have shown notable success in short-term predictions by minimizing temporal mean squared errors (MSEs). Prominent methods include recurrent neural networks (RNNs) (Madondo & Gibbons, 2018; Dudukcu et al., 2023), long short-term memory networks (LSTMs) (Sangiorgio & Dercole, 2020; Langeroudi et al., 2022), and reservoir computing methods (Pathak et al., 2018; Yan et al., 2024). These approaches have been applied to classic 1-D examples, such as the Lorenz 63, Lorenz 96, and Kuramoto-Sivashinsky equations. However, due to the inherent instability of chaotic systems, these methods often experience exponential error accumulation over time.

To address this issue, recent advances leverage ergodicity, a key property of many chaotic systems on strange attractors (Eckmann & Ruelle, 1985; Young, 2002), to stabilize the method performance in long-term predictions by aligning predicted distributions with ground-truth distributions. Two notable approaches utilize optimal transport theory (Jiang et al., 2024; Schiff et al., 2024) to improve long-term prediction accuracy, i.e. incorporating a regularized transport term to penalize discrepancies between the predicted and

true distributions. However, matching distributions often encounters the curse of dimensionality (Kloeckner, 2020; Zhang et al., 2023; Vögler, 2023), particularly when chaotic systems are high-dimensional and characterized by multimodal probability distributions (Sengupta, 2003; Zelik, 2022). High-dimensional chaotic systems, such as turbulent flows, typically manifest in phase spaces with dimensions exceeding $\mathcal{O}(10^5)$ and exhibit non-Gaussian statistical properties (Biferale et al., 2006). The limitation of distribution-matching methods lies in their computational intractability for such high-dimensional spaces, compounded by the difficulty of accurately capturing the intricate statistical structures of non-Gaussian distributions (Korotin et al., 2021).

**Operator-based models.** Rooted in operator theory, operator-based approaches offer a promising alternative for modeling dynamic systems (Marchenko, 2012). Unlike sequence models, these methods do not require prior knowledge of the underlying systems. Instead, they leverage function approximation theory to identify an operator that captures the system's evolution within a predefined function space (Smale & Zhou, 2007). Two representative methods are transfer operator (Jørgensen, 2001) and Koopman operator (adjoint form of transfer operator) (Brunton et al., 2017; Cheng et al., 2023; Brunton et al., 2021; Mezić, 2021; Cheng et al., 2025). The transfer operator models the evolution of probability density functions, while the Koopman operator focuses on the evolution of observable functions. Classical transfer and Koopman operators leverage kernel methods within the framework of reproducing kernel Hilbert space or Banach space (Klus et al., 2020; Ikeda et al., 2022; Hou et al., 2023; Yang et al., 2025). These approaches face computational bottlenecks due to the prohibitive cost of matrix inversion (Bousquet & Herrmann, 2002) for high-dimensional chaotic systems. Consequently, applying kernel-based techniques to such systems remains a significant challenge. In recent years, deep learning techniques have been integrated with operator theory and have yielded significant advancements in solving partial differential equations (PDEs). Methods such as Deep Operator Network (DeepONet) (Lu et al., 2021) and Fourier Neural Operator (FNO) (Li et al., 2020) have shown their effectiveness as approximators for initial value problems in PDEs. Many variants of these two methods have been developed to solve large-scaled PDEs, such as U-shaped FNO (Rahman et al., 2022), multiwavelet-based operator (Gupta et al., 2021). However, chaotic systems are highly sensitive to initial values (Kolesov & Rozov, 2009), purely predicting the long-term behavior based on learned operators cannot guarantee long-term stability. Hence, a more problem-specific framework is necessary to model chaotic systems. Markov Neural Operator (MNO) (Li et al., 2022a) as a closely related framework, introduced hard constraints upon the architecture of FNO to enhance long-term pre-dictions for dissipative chaotic systems by enforcing the forward invariance via the constraint. Yet no loss function in MNO theoretically guarantees the ergodicity for a long-term prediction, leaving potential discrepancies in statistical properties unresolved.

**Transformer-based models.** Leveraging the flexibility and universality of transformers, researchers started to integrate sequence models and operator theory with transformer architectures to model complex dynamical systems (Li et al., 2022b; Guibas et al., 2021; Hao et al., 2023; Kissas et al., 2022; Zhang & Gilpin, 2024). For example, the Fourier Transformer has been proposed as an extension of the Fourier Neural Operator to solve a variety of PDEs (Li et al., 2022b; Guibas et al., 2021; Hao et al., 2023; Chen et al., 2024). To reduce the computational cost of standard scaled-dot product attention, several pioneering works have eliminated the Softmax operation and utilized matrix associativity to achieve linear complexity attention, which has shown promise for large-scale PDE modeling (Cao, 2021; Li et al., 2024). Despite the significant advancements in transformer-based methods for dynamical systems, most efforts have primarily concentrated on general PDEs. This leaves a notable research gap: the lack of a well-designed transformer architecture tailored specifically for large-scale chaotic systems.

To address this gap, we first propose a tailored transformer-based model for predicting chaotic systems. Our key contributions can be summarized as follows:

- Designed for modeling chaotic systems and aligned with their intrinsic properties, we introduce A3M modifications into factorized attention (Figure 1(b)) to re-design the scaled-dot product attention, and enhance it with random Fourier positional encoding for improved representation of complex chaos patterns.

- Building on the Koopman-Neumann ergodic theorem (Neumann, 1932; Mezić, 2021), we propose a novel loss function with a unitary constraint on the forward operator (Figure 1(c)), which is scalable to capture long-term statistics of large-scale chaotic systems and ensure stability without directly matching probability distributions.

- We introduce datasets of two high-dimensional chaotic systems: 1) turbulent channel flow with $140k$ snapshots in the 3D simulation; 2) the Kolmogorov Flow simulation of $185k$ 2D vorticity states. Both datasets with details in Appendix D and G have been carefully processed to ensure consistency and usability, making them well-suited for early-stage machine learning research on ergodic chaotic systems.

We benchmark our algorithm with the state-of-the-art operator-based and transformer-based methods on two challenging chaotic systems. The results consistently demonstrate superior short-term prediction accuracy and long-term statistical consistency in high resolution. Our core framework is illustrated in Figure 1.

## 2. Preliminary

**Notation.** $(X, \mathcal{B}, \mu)$ denotes the measure space, with set $X$, Borel $\sigma$-algebra $\mathcal{B}$ and measure $\mu$. The forward map on the state space is denoted as $T$. The Lebesgue space with the $p$-norm is denoted as $\mathcal{L}^p(X, \mu)$, abbreviated as $\mathcal{L}^p$ in this paper. In particular, the Lebesgue space $\mathcal{L}^2$ is equipped with an inner product structure as $\langle \cdot, \cdot \rangle$. Feature functions $\phi, \psi$ belong to the $\mathcal{L}^2$ space. $\square^T$ denotes the transpose of a real operator, and $\square^*$ denotes the conjugate transpose of a complex operator. The symbol $z_j^i$ indicates the $i$-th indexed state at the $j$-th time step, and $\xi_i$ represents the coordinate of the $i$-th index.

**Problem formulation.** The objective is to forecast the behavior of ergodic chaotic systems, described as

$$z_{k+1} = T(z_k), \quad z_k \in \mathcal{M}, \tag{1}$$

where $T : \mathcal{M} \to \mathcal{M}$ is a nonlinear forward map on a compact set $\mathcal{M} \subseteq \mathbb{R}^{S_1 \times S_2 \times \cdots \times S_M}$. The state $z_k$ represents physical quantities defined on a uniform grid with dimensions $S_1, S_2, \ldots, S_M$, with the total number of grid points $N = S_1 \times S_2 \times \cdots \times S_M$. Each component of $z_k$ is bounded and represents a physical quantity at the $k$th grid point.

**Definition 2.1** (Measure-Preserving Transformation and Ergodic (Cornfeld et al., 2012)). Let $(X, \mathcal{B}, \mu, T)$ define a measure-preserving transformation (MPT), where for every $E \in \mathcal{B}$, the measure satisfies $\mu(T^{-1}E) = \mu(E)$. An MPT is called ergodic if, for any invariant set $E$, either $\mu(E) = 0$ or $\mu(X \setminus E) = 0$. In this case, $\mu$ is referred to as an ergodic measure.

Intuitively, ergodicity ensures spatial statistics are compatible with temporal statistics. This implies that the distribution of system trajectories follows an invariant measure that remains consistent over time. Many chaotic systems exhibit this property on their attractors, such as Lorenz systems (Shi et al., 2020), Kuramoto-Sivashinsky equation (Yang, 2006) and turbulent fluids (Galanti & Tsinober, 2004; Hairer & Mattingly, 2006). By preserving ergodicity, the learned model can capture the invariant statistical behaviors of the attractors.

Learning the forward map $T$ of chaotic systems while preserving ergodicity is challenging. To address this, our model adopts an operator to forward in feature space:

$$\phi(z_{k+1}) = (\mathcal{G}\phi)(z_k), \tag{2}$$

where $\phi$ is the feature map as $\phi : \mathcal{M} \to \mathbb{R}$, with $\phi \in \mathcal{F} \subset \mathcal{L}^2$ and function space $\mathcal{F}$ on $\mathcal{M}$. The operator $\mathcal{G}$ describes the evolution of learnable features. The time evolution of these features can be obtained by iteratively applying $\mathcal{G}$ in $\mathcal{L}^2$ space. By applying a decoding function $\phi^\dagger$, the state $z_{k+1}$ can be reconstructed. The choice of $\mathcal{L}^2$ space is motivated by the well-developed theory of ergodicity, stemming from the contributions of Von Neumann (Neumann, 1932).

**Theorem 2.2** (Von Neumann Ergodic Theorem (Neumann, 1932)). *Let* $(X, \mathcal{B}, \mu, T)$ *be an MPT. If* $\phi \in \mathcal{L}^2$, *then* $\lim_{N \to \infty} \frac{1}{N} \sum_{k=0}^{N-1} \mathcal{G}^k \phi = \overline{\phi}$ *where* $\overline{\phi}$ *is invariant. If* $T$ *is ergodic, then* $\overline{\phi} = \int \phi d\mu$.

This theorem demonstrates that for an ergodic measure-preserving transformation (MPT) $T$, the associated operator $\mathcal{G}$ is unitary and preserves the norms in the $\mathcal{L}^2$ space. This implies that the time averages of the $\mathcal{L}^2$ functions converge to their space averages, ensuring the long-term predictability of chaotic systems. By combining with Equation 2, the operator $\mathcal{G}$ preserves the ergodic property during long-term predictions. Our objectives are twofold: (1) to achieve accurate short-term predictions and (2) to capture statistical properties by preserving ergodicity in the $\mathcal{L}^2$ space.

## 3. Method

In this section, we present our customized transformer architecture designed for modeling ergodic chaotic systems. This method is structured into three key components: 1) introducing an attention mechanism to identify spatial correlations based on random Fourier features, which naturally align with chaotic system properties as discussed in section 3.1, 2) efficient processing of multi-dimensional tensors with capturing extreme values by factorizing attention and the proposed A3M block, detailed in section 3.2), and 3) an efficient constraint for learning the unitary operator in the high-dimensional feature space to preserve ergodicity using Hutchinson's stochastic trace estimation with details in section 3.3).

### 3.1. Attention Mechanism with Random Fourier

Give three sets of feature functions (or vectors), namely the queries $\{q^i\}_{i=1}^N$, keys $\{k^i\}_{i=1}^N$, and values $\{v^i\}_{i=1}^N$, the self-attention mechanism (Vaswani, 2017; Han et al., 2022) calculates a weighted average of values $\phi(z^i) = \sum_{j=1}^N h(q^i, k^i)v^j$, where $q^i, k^i, v^i \in \mathbb{R}^d$. The attention weights $h(\cdot, \cdot)$ are defined as the scaled-dot product with a Softmax function in the original transformer (Vaswani, 2017): $h(q^i, k^j) = \frac{\exp(\langle q^i, k^j \rangle / \sqrt{d})}{\sum_{s=1}^N \exp(\langle q^i, k^s \rangle / \sqrt{d})}$. It is interesting to interpret the weighted average $\phi(z^i) \in \mathcal{L}^2$ as a learnable feature of state $z^i$ as shown in Equation (2).

The queries/keys/values are typically obtained through learn-

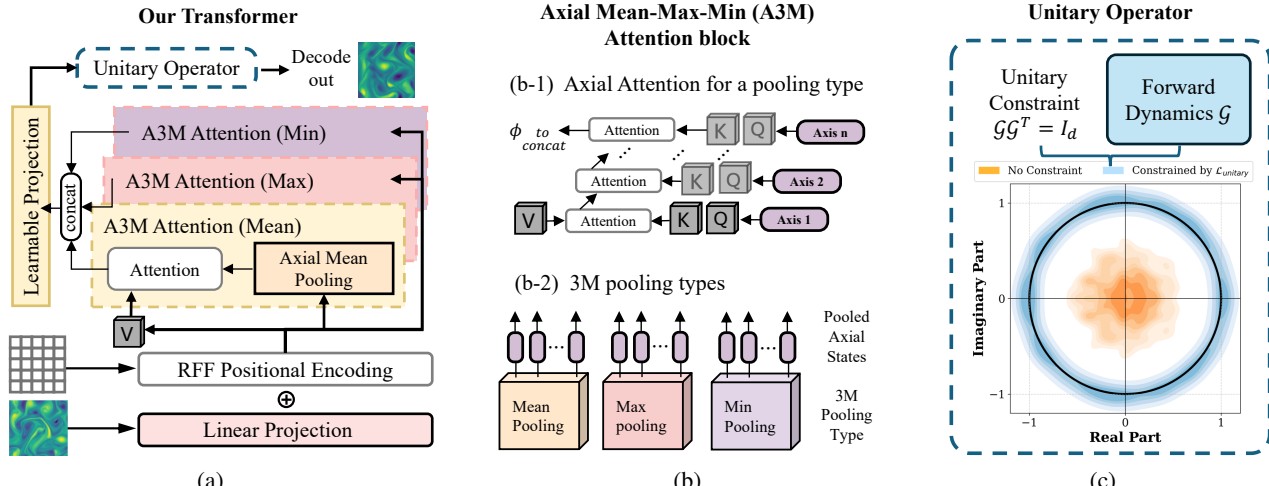

**Figure 1.** The framework of ChaosMeetsAttention. Figure 1(a) presents the main structure of our transformer, highlighting the modified positional encoding, attention mechanism, and unitary operator. Figure 1(b) illustrates the efficient Axial Mean-Max-Min (A3M) attention, designed to capture both statistical moments and extreme values in the physical field. Finally, Figure 1(c) aligns with the Von Neumann mean ergodic theorem described in 2.2, demonstrating how the unitary operator preserves statistical properties.

able linear projections:

$$q^i = z^i W_q, \qquad k^i = z^i W_k, \qquad v_i = z^i W_v, \quad (3)$$

where $z^i \in \mathbb{R}^{d_{in}}$ represent the $i$-th input physical quantity with dimensions $d_{in}$ in the domain $\mathcal{M}$, and $\{W_q, W_k, W_v\}$ are learnable projection matrices. Standard self-attention mechanisms do not explicitly encode spatial correlations. They compute similarity in the learned feature space and apply probability-weighted averaging via the Softmax operation. This formulation lacks an explicit spatial distance representation. To overcome this limitation, we introduce a distance-based Gaussian kernel to capture spatial correlations based on relative distances. This kernel function decays smoothly with increasing distance, aligning with the spatial mixing behaviors observed in chaotic systems.

To efficiently compute this kernel, we employ random Fourier features (Rahimi & Recht, 2007; Tancik et al., 2020) for the approximation of a Gaussian kernel using randomly sampled Fourier basis functions. This distance-based formulation shares an interesting connection with the topological mixing property of chaotic systems (Xiong & Yang, 1991). In chaotic systems, *topological mixing describes how any localized region of the state space eventually spreads and overlaps with other regions, allowing local perturbations to propagate throughout the space.* Similarly, the Gaussian kernel-based attention mechanism enables each point to interact with its surroundings with a strength that smoothly decays with distance, providing a natural way to model how information or influence propagates through space in physical fields. The kernel's bandwidth controls the characteristic length scale of these spatial interactions, analogous

to how mixing rates in physical systems determine the scale of spatial correlations.

**RFF Positional Encoding.** Based on Bochner's theorem (Bochner et al., 1959), random Fourier features (RFFs) can be used to approximate a stationary kernel. For simplicity, we demonstrate the applications of RFFs in a 1-dimensional grid domain, which can be easily extended to higher-dimensional grid domains. We use a random Fourier map $\theta$ to featurize input grids, which project input coordinate index $i$ to the surface of a sphere with a set of sinusoids: $\theta(i) = [\frac{1}{\sqrt{m}}cos(2\pi B\xi_i), \frac{1}{\sqrt{m}}sin(2\pi B\xi_i)]$, where $\xi_i$ is the coordinate of $i$-th index, $B \in \mathbb{R}^m$ with $B_i \sim \mathcal{N}(0, \sigma^2)$ for $i = 1, ..., m$. From trigonometric identities, the inner product of $\theta(i)$ and $\theta(j)$ is formulated as:

$$
\begin{aligned}
\theta(i)^T \theta(j) &= \theta(i - j) \\
&= \lim_{m \to \infty} \frac{1}{m} \sum_{j=1}^{m} cos(2\pi b_j(\xi_i - \xi_j)) \quad (4) \\
&\approx \exp\big(-\pi\sigma^2 \|\xi_i - \xi_j\|_2^2\big).
\end{aligned}
$$

The last line of Equation (4) is from the characteristic function of Gaussian distribution (Shiryaev, 2016) with details in Appendix B. By constructing the attention mechanism with RFF, the corresponding queries and keys become $g(q^i) = q^i\theta(i)$ and $g(k^j) = k^j\theta(j)$. Let $Q = [g(q^1), \cdots, g(q^N)]$, $K = [g(k^1), \cdots, g(k^N)]$ and $V = [v^1, \cdots, v^N]$. The fea-

ture $\phi(z^i)$ is expressed as:

$$
\begin{aligned}
\phi(z^i) &= g(q^i)K^TV \\
&= \sum_{s=1}^{N} \langle g(q^i), g(k^s) \rangle v^s \\
&= \sum_{s=1}^{N} \theta(i-s)\langle q^i, k^s \rangle v^s \\
&\approx \sum_{s=1}^{N} \exp\left(-\pi\sigma^2\|\xi_i - \xi_s\|_2^2\right)\langle q^i, k^s \rangle v^s.
\end{aligned}
\tag{5}
$$

Here, $\phi(z^i)$ represents a quadrature of the integral kernel, capturing the weighted average of values. The spatial correlation between the $i$-th index and other indices $s$ decays according to the Gaussian kernel, leading to stronger influence from nearby points, aligning with the topological mixing property (see section 3.1). The parameter $\sigma$ in Equation (5) controls the decay rate of these spatial correlations: a smaller $\sigma$ leads to slower decay and potential underfitting, while larger $\sigma$ causes rapid decay, increasing the risk of overfitting. An ablation study in section 4.3 examines the impact of different $\sigma$ values on model performance.

## 3.2. Factorizing as Multi-Dimensional Tensor

The computation complexity of the attention mechanism increases quadratically with the total domain grid points $N$. Unlike natural language processing (NLP) sequences, the domain $\mathcal{M}$ has a uniform grid structure as $S_1 \times S_2 \times \cdots \times S_M$. Directly applying attention as in Equation (5) requires $S_1 \times S_2 \times \cdots \times S_M$ times matrix multiplications, which becomes prohibitively expensive as the grid resolution increases. To address this, we leverage a tensor factorization approach from (Li et al., 2024), which reduces the number of matrix multiplications from $S_1 \times S_2 \times \cdots \times S_M$ to $S_1 + S_2 + \cdots + S_M$. The basic idea is to factorize the attention operation along each axis of the grid domain, resulting in a linear increase in computational complexity with grid resolution rather than a quadratic or cubic increase.

**Multi-dimensional tensor product.** The data in the grid domain can be represented as a tensor $A \in \mathbb{R}^{S_1 \times S_2 \times \cdots \times S_M}$. The product with a matrix $W \in \mathbb{R}^{J \times S_m}$ on the $m$-th mode in a tensor of shape $S_1 \times S_2 \times \cdots \times J \times S_M$, is defined as:

$$
(A \times_m W)_{i_1 i_2 \cdots j \cdots} = \sum_{i_m}^{S_m} A_{i_1 i_2 \cdots j \cdots i_M} W_{j i_m}. \tag{6}
$$

According to the principles of multi-dimensional tensor, the attention mechanism with positional encoding along each axis can be expressed as:

$$
\phi(z^i) = \int_{\Xi^M} \cdots \int_{\Xi^1} \sum_{s=1}^{S_1} \langle g^1(q^i), g^1(k^s) \rangle v^s d\xi^{(1)} \cdots d\xi^M,
\tag{7}
$$

where $\Xi^i$ denotes the $i$-th coordinate axis of grids. Here, $g^j(q^i)$ and $g^j(k^i)$ for $j = 1, ..., M$ and $i = 1, ..., S_j$ are pooling from multi-dimensional tensors $Q, K \in \mathbb{R}^{S_1 \times S_2 \times \cdots \times S_M \times d}$ along all dimensions except the $j$-th dimension, e.g.,

$$
g^j(q^i) = \frac{1}{\prod_{m \neq j} S_m} \sum_{\substack{i_1, ..., i_M \\ i_m \neq i_j}} Q_{i_1, i_2, ..., i_M, i}
$$

with mean aggregation. Different from (Li et al., 2024), we propose separate attention heads in the transformer block to apply axial attention with mean, max, and min poolings, collectively named the *A3M Attention blocks* (as shown in Figure 1(b)). This design captures both statistical moments and extreme values in the field (Yeung et al., 2015; Farazmand & Sapsis, 2017; Sapsis, 2021), which are critical for chaotic systems like turbulent flows, where both statistical properties and local extremas significantly influence dynamics. By recursively calculating the quadrature of the integral kernel along each axis from 1 to $M$, the number of required matrix multiplications is reduced to $S_1 + S_2 + \cdots + S_M$, achieving a considerable improvement in computational efficiency. Figure 1(b) illustrates the proposed attention mechanism.

## 3.3. Embedding Ergodicity on Transformer

By connecting chaos to operator theory and Von Neumann's statement in Theorem 2.2, it provides a framework for preserving the statistical behavior of chaotic systems through the operator $\mathcal{G}$. Specifically, the eigenvalue (or spectrum) of the operator $\mathcal{G}_m$ lies on the complex unit circle $Eigen(\mathcal{G}) \subset \{r \in \mathbb{C} \mid |r| = 1\}$ (Avigad, 2009).

To identify the unitary operator in the ergodic state, we leverage the intrinsic property that $\mathcal{G}$ and its conjugate transpose $\mathcal{G}^*$ satisfy $\mathcal{G}^*\mathcal{G} = I_d$. This property confirms that $\mathcal{G}$ is unitary, preserving both norms and inner products, and establishes a well-defined backward dynamic. Specifically, the backward operator $\mathcal{G}^{-1}$ is equivalent to $\mathcal{G}^*$. The conjugate transpose relationship $\mathcal{G}^*\mathcal{G} = \mathcal{G}\mathcal{G}^* = I_d$ ensures that $\mathcal{G}$ is invertible, with its inversion fully consistent with unitary, as shown in Figure 1(c).

To incorporate the theorem in practice, we introduce a regularized term to penalize it, and the loss function is expressed as:

$$
\arg\min_{\hat{\mathcal{G}}} \mathbb{E}_{z \sim \mu}\left[\|\hat{\mathcal{G}}\phi(z_k) - \phi(z_{k+1})\|_2 + \lambda\mathcal{L}_{unitary}(\hat{\mathcal{G}})\right],
\tag{8}
$$

where the first term represents the forward loss, and $\mathcal{L}_{unitary}(\hat{\mathcal{G}})$ is the unitary regularization with coefficient $\lambda \in (0, 1]$. To enhance the scalability of the framework, we propose a novel approach by incorporating Hutchinson's stochastic trace estimator.

To express the unitary operator on the torus, as discussed in (Das, 2023), it can be represented as the form $\underbrace{U(1) \times U(1) \times \cdots \times U(1)}_{d/2}$, where $U(1) = \{\exp(i\theta) \mid \theta \in [0, 2\pi]\}$ is the circle group. It is well-known that $SO(2)$ is isomorphic to $U(1)$. Since the parametrization of neural networks is typically in the real space instead of complex space, the real representation of product circle group $U(1) \times U(1) \times \cdots \times U(1)$ corresponding to $\underbrace{SO(2) \times SO(2) \times \cdots \times SO(2)}_{d/2}$ can be embedded into $SO(d)$, where the parametrized dimension $d$ is an even number. Here, $SO(d) = \{A \in \mathbb{R}^{d \times d} \mid AA^T = A^T A = I_d, det(A) = 1\}$ denotes the $d$-dimensional special orthogonal group. In such a situation, $\mathcal{L}_{unitary}(\hat{\mathcal{G}})$ can be defined as:

$$\mathcal{L}_{unitary}(\hat{\mathcal{G}}) := \mathbb{E}_{v \sim \text{Unif}(S^{d-1})} \left[ \left| v^T \hat{\mathcal{G}}^T \hat{\mathcal{G}} v - 1 \right| \right]. \quad (9)$$

Here, $v$ is a random unit vector sampled uniformly from the unit $(d-1)$-sphere $S^{d-1} \subset \mathbb{R}^d$, i.e., $\|v\|_2 = 1$. The regularized term $\mathcal{L}_{unitary}(\hat{\mathcal{G}})$ constrains $\hat{\mathcal{G}}$ within $SO(d)$. The sample number of the unit random vector is $k$, and the upper error bound of the stochastic trace estimator is scaling as $\mathcal{O}(\frac{1}{\sqrt{k}})$ (Meyer et al., 2021). The number of sample $k$ is proportional to $\frac{\log(1/\delta)}{\epsilon^2}$, where $\epsilon$ measures the upper error bound with at least $1 - \delta$ probability. We impose a soft constraint on our loss function, making our learned physics align with Von Neumann's mean ergodic theorem in 2.2.

Compared to the penalization in (Cheng et al., 2025), our proposed loss function demonstrates higher efficiency when applied to large-scale chaotic systems than using Frobenius-norm regularized term (Golub & Van Loan, 2013). The computational complexity of the Frobenius-norm regularized term is $\mathcal{O}(d^3 + 2d^2)$, whereas the complexity of the stochastic trace estimator is $\mathcal{O}(kd^2)$. Given that $d$ is typically large for approximating evolution in $\mathcal{L}^2$ space, our method provides a more computationally efficient approach.

---

**Algorithm 1** $\mathcal{L}_{unitary}$ with Hutchinson's Stochastic Trace Estimator

---

**Require:** Operator $\hat{\mathcal{G}} \in \mathbb{R}^{d \times d}$, batch size $B$
 1: Initialize $\mathcal{L}_{unitary} = 0$
 2: **for** $b = 1$ to $B$ **do**
 3:   Sample $v^{(b)} \sim \text{Unif}(S^{d-1})$
 4:   $q^{(b)} = v^{(b)^T} \hat{\mathcal{G}}^T \hat{\mathcal{G}} v^{(b)}$
 5:   $\mathcal{L}_{unitary} \leftarrow \mathcal{L}_{unitary} + |q^{(b)} - 1|/B$
 6: **end for**
 7: **return** $\mathcal{L}_{unitary}$

---

## 4. Numerical Experiments

In this section, we comprehensively evaluate the performance of our model on two high-resolution chaotic systems. Kolmogorov flow has external periodic forcing and structured large-scale dynamics, while turbulent channel flow involves flow-driven turbulence with more complex, anisotropic behavior. The experiments are repeated with three random seeds, and the mean value is reported in the results tables and figures in section 4.1, 4.2 and Appendix F. We also include details of computational cost to demonstrate fair comparison with baselines and the efficiency our methods on increasing state resolutions in Table 7 and 8.

**Baselines.** We compare our transformer with four competitive baselines covering different learning mechanisms in dynamical systems, including: (a) Markov neural operator (**MNO**) (Li et al., 2022a), an FNO-based model designed to capture long-term dissipative chaotic behavior in sequential data by incorporating a tailored constraint; (b) **UNO** (Rahman et al., 2022), a U-shaped FNO architecture that incorporates skip connections; (c) Multiwavelet-based Operator (**MWT**) (Gupta et al., 2021) a neural operator using wavelet transformations to effectively model dynamical behaviors; and (d) **Factformer** (Li et al., 2024), an attention-based model for learning PDEs, designed to factorize attention computations along different spatial axes to achieve scalability. With the specialized design, MNO has demonstrated state-of-the-art performance in capturing the long-term statistics of chaotic systems.

**Benchmarks.** We choose two turbulent fluid dynamics as popular benchmarks for learning states from chaotic systems: Kolmogorov Flow (**KF256**): a 2D shear flow characterized by sinusoidal velocity fields in one direction and external forcing in the perpendicular direction. In our experiments, we utilize a 256×256 resolution vorticity field to study the short- and long-term performance (He, 2025). Turbulent Channel Flow (**TCF**): a 3D canonical wall-bounded flow where turbulence develops between two parallel planes due to a pressure gradient, characterized by turbulent velocity fields, enhanced mixing, and distinct near-wall and core regions. We extract the 3-channel 2D velocity field from the 3D simulation and evaluate our approach with baselines on the resolution $192 \times 192$ over all channels (He et al., 2025). We generated and prepared these datasets with details in Appendix D and G.

**Evaluations.** We evaluate the models in terms of short- and long-term performances. For short-term performance, we use the relative norm $L^2$, following (Li et al., 2022a). Due to the high dimensionality of our experiments, achieving robust estimations of long-term statistics is challenging. To ensure a valid comparison, we assess long-term perfor-

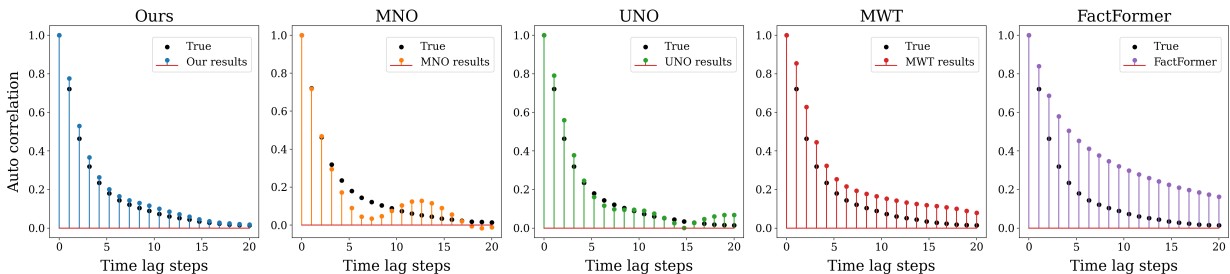

*Figure 2.* Time correlation of our long-term predictions of TCF and the baselines.

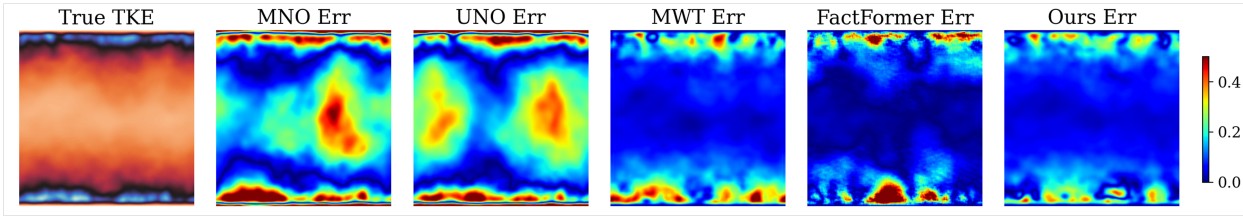

*Figure 3.* TKE error of long-term predictions of TCF. The left-most snapshot indicates the TKE of ground truth trajectories; and the rest snapshots present the absolute percentage error of TKE in 1000-step rollout predictions from baselines and our approach.

mance using three distinct metrics: (1) the mean energy absolute percentage error (ME-APE), which quantifies the absolute deviation of predicted energy in each frequency mode from the ground truth; (2) a weighted variant (ME-LRw), which assigns higher weights to modes with greater energy, emphasizing low-frequency components following (Schiff et al., 2024); and (3) the absolute difference in estimated mixing rates ($\Delta\lambda$) between true and generated trajectories, a key indicator of chaotic systems that reflects how quickly the system loses memory of initial conditions. The mixing rate is estimated from time correlation function, which we also compare in the KF256 and TCF to analyze the detailed temporal decay. Meanwhile, we also compare the absolute difference of the estimated Turbulent Kinetic Energy (TKE), which describes the turbulent energy distribution of the underlying dynamical system; and Kullback–Leibler divergence (KLD) on principle components, which measures the statistical distance between the estimated distributions from the generated data and the true data. More details on the evaluation metrics are provided in Appendix C and E.

## 4.1. Kolmogorov Flow

Table 1 demonstrates the better performance of our transformer model across both short- and long-term metrics tasks on the KF256 dataset. In the short-term evaluation, our model achieves the lowest relative $L^2$ error, with a 6.52% and 1.55% improvement over the best baseline for $\tau \in \{5, 25\}$ steps rollout. For long-term statistics, our approach consistently outperforms existing methods in all metrics, achieving notable improvements of 23.1%, 21.0%,

and 20.0% in matching the energy spectrum according to ME-APE, ME-LRw, and the mixing rate $\Delta\lambda$, respectively. These results highlight the effectiveness of our method on both short- and long-forecasting horizons.

*Table 1.* Short-term and long-term performance of baselines and our transformer prediction on KF256.

| Modules | Short-term (Rel-$L^2$) | | Long-term | | | |
|---|---|---|---|---|---|---|
| | $\tau = 5$ | $\tau = 25$ | ME-APE | ME-LRw | $\Delta\lambda$ | KLD |
| MNO | 1.02 | 1.29 | 0.42 | 0.70 | 0.40 | 0.42 |
| UNO | 0.92 | 1.32 | 0.22 | 0.36 | 0.11 | 0.38 |
| MWT | 0.95 | 1.32 | 0.26 | 0.39 | 0.17 | 0.40 |
| FactFormer | 0.97 | 1.35 | 0.13 | 0.19 | 0.10 | 0.37 |
| Ours | **0.86** | **1.27** | **0.10** | **0.15** | **0.08** | **0.29** |
| Advantage (%) | 6.52 | 1.55 | 23.1 | 21.0 | 20.0 | 21.6 |

Detailed results of time correlation to derive the mixing rate and the TKE absolute errors are provided in Appendix F.1, where Figure 5 demonstrates our model captures the correlation decay of the underlying system, and Figure 6 further validates the advantage of applying A3M in capturing extreme values and the unitary operator to maintain invariant statistics.

## 4.2. Turbulent Channel Flow

From the results in Table 2, our method achieves consistently better performance for both short- and long-term predictions. For short-term metrics, we achieve the lowest relative $L^2$-errors, with a 25.0% and 7.14% improvement over baselines at autoregressive forward steps $\tau = 5$ and $\tau = 25$, respectively, demonstrating improved accuracy in

short-term dynamics. For metrics evaluating energy spectrum accuracy (ME-APE, ME-LRw) and the mixing rate error ( $\Delta\lambda$ ), our model outperforms all baselines by 30.8%, 25.0%, and 27.3%, respectively. These gains highlight our model's ability to maintain precise statistical and physical consistency over long extended horizons.

*Table 2.* Short-term and long-term prediction performance of baselines and our transformer predictions for TCF.

| Modules | Short-term (Rel-$L^2$) | | Long-term | | | |
|---|---|---|---|---|---|---|
| | $\tau = 5$ | $\tau = 25$ | ME-APE | ME-LRw | $\Delta\lambda$ | KLD |
| MNO | 0.08 | 0.24 | 0.84 | 1.95 | 0.11 | 3.73 |
| UNO | 0.07 | 0.19 | 0.86 | 2.07 | 0.12 | 4.89 |
| MWT | 0.04 | 0.14 | 0.21 | 0.19 | 0.11 | 2.70 |
| FactFormer | 0.09 | 0.17 | 0.13 | 0.12 | 0.23 | 3.34 |
| Ours | **0.03** | **0.13** | **0.09** | **0.09** | **0.08** | **1.97** |
| Advantage (%) | **25.0** | **7.14** | **30.8** | **25.0** | **27.3** | **27.0** |

Moreover, Figure 2 demonstrates a more consistent mixing rate with the ground truth, characterized by a monotonic decrease in correlation. In contrast, models such as MNO and UNO, which are based on the integer Fourier spectrum, exhibit a more periodic correlation pattern. This may contract with the mixing property of chaotic systems. We also compare the accuracy of TKE in Figure 3. The top and bottom boundaries of all other models exhibit significant errors, whereas our model demonstrates more stable performance across the entire domain, which is attributed to the unitary-constrained operator, preserving the invariant statistics.

### 4.3. Ablation Study

We revisit the Kolmogorov Flow system in a lower resolution of $128 \times 128$ as KF128, setting training and evaluation random seed as 0, to conduct extensive ablation studies to validate the design of our approach efficiently.

Firstly, we evaluate the effectiveness of the unitary operator and the three-channel pooling method as key modules of our transformer. We compare three model versions as follows: (1) the basic version, which is composed of one Mean-pooling dimension reduction module, four factorized attention blocks, and one forward operator learned by a simple neural network; (2) an introduction of a Min- and a Max-pooling module to the basic version; (3) a further unitary-constrained version introduced with $\mathcal{L}_{unitary}$ on the operator. Table 3 includes the metrics evaluated for each configuration, providing insight into how these modules affect short- and long-term performance.

The introduction of a unitary-constrained operator and the Min- and Max-pooling module consistently enhances the modeling performance of ergodic chaotic systems across both short- and long-term metrics. According to the Von Neumann mean ergodic theorem, the unitary operator en-

*Table 3.* Short-term and long-term evaluation on KF128 of key modules in our transformer.

| Modules | Short-term (Rel-$L^2$) | | Long-term | | |
|---|---|---|---|---|---|
| | $\tau = 5$ | $\tau = 25$ | ME-APE | ME-LRw | $\Delta\lambda$ |
| Base | 1.02 | 1.43 | 0.17 | 0.23 | 0.11 |
| + A3M Att. | 0.98 | 1.38 | 0.15 | 0.21 | 0.10 |
| + Unitary Op. | **0.92** | **1.30** | **0.14** | **0.19** | **0.08** |
| Advantage (%) | **6.12** | **5.80** | **6.67** | **9.52** | **20.0** |

sures the preservation of long-term statistics in the $\mathcal{L}^2$ space. This property is reflected in the improved long-term metrics presented in Table 3. Additionally, the A3M pooling effectively captures local extreme values, contributing to improvements in both short- and long-term performance.

Secondly, we investigate the effect of RFF positional encoding on our transformer model by examining the influence of the hyperparameter $\sigma$, as described in (Tancik et al., 2020). Specifically, we explore values of $\sigma$ in the range $[1, 4, 8, 16, 32]$. The kernel bandwidth $\sigma$ determines the characteristic length scale of spatial interactions, akin to how mixing rates in physical systems influence the scale of spatial correlations. When the bandwidth is large, spatial interactions are highly localized, making each point nearly self-determined and ignoring spatial correlation from other positions. Conversely, a small bandwidth results in almost uniform interactions, representing a state of maximal mixing. From the evaluation results shown in Table 4, we observed that for $\sigma \geq 8$, the model more effectively captured high-frequency spectrums, counting on more localized interactions. In contrast, for $\sigma \in \{1, 4\}$, the model exhibited better performance in capturing low-frequency spectrums. This phenomenon aligns with theoretical insights discussed in (Tancik et al., 2020). From the attention maps in Figure 8, the results also indicate that a small $\sigma$ produces overly smoothed attention that leads to underfitting; while large values of $\sigma = 32$ produce noisy attention and excessively large $\sigma$ may lead to overfitting.

*Table 4.* Short-term and long-term evaluation on KF128 of Kernel Bandwidth in positional encoding.

| Bandwidth | Short-term (Rel-$L^2$) | | Long-term | | |
|---|---|---|---|---|---|
| | $\tau = 5$ | $\tau = 25$ | ME-APE | ME-LRw | $\Delta\lambda$ |
| $\sigma = 1$ | 0.98 | 1.33 | 0.22 | 0.27 | 0.17 |
| $\sigma = 4$ | 0.94 | 1.33 | 0.19 | 0.21 | 0.08 |
| $\sigma = 8$ | **0.92** | **1.30** | **0.14** | **0.19** | 0.08 |
| $\sigma = 16$ | **0.92** | **1.30** | 0.15 | **0.19** | **0.07** |
| $\sigma = 32$ | 0.93 | **1.30** | **0.14** | 0.20 | 0.11 |

## 5. Conclusion

The autoregressive generation of long-term trajectories for large-scale chaotic systems remains a significant challenge

in machine learning. We propose a novel transformer-based framework specifically designed for large-scale ergodic chaotic systems. Our approach enhances the transformer architecture by redesigning the attention blocks with A3M pooling in factorized attention to learn the topology mixing property and capture the extreme behavior, leading to improved long-term trajectory generation. Moreover, we introduce a novel loss function inspired by the classical von Neumann ergodic theorem, ensuring the preservation of long-term statistical properties in chaotic systems. Experimental results show that the proposed model achieves superior performance over both operator-based and transformer-based methods and excels in both short-term accuracy and the preservation of long-term statistical properties, achieving relative 23.1% gain among short- and long-term metrics on the high-resolution KF256 dataset and 30.8% on the challenging 3-channel high-resolution TCF dataset.

**Limitations and future work.** Our approach currently focuses on ergodic chaotic systems and does not generalise well to non-ergodic system, where long-term statistical properties are not well-defined. Future research could explore and extend the framework to accommodate such cases. Additionally, our method relies on uniform grid structures, which may limit its applicability to more complex chaotic systems. Future work could explore mesh-free techniques, such as graph-based transformers, to model chaotic physical fields without relying on grid-based data structures.

## Acknowledgments

The authors thank the reviewers and the program committee of ICML 2025 for their insightful feedback and constructive suggestions that have helped improve this work. We also acknowledge the support from the University College London (Dean's Prize, Chadwick Scholarship), the Engineering and Physical Sciences Research Council projects (EP/W007762/1), the United Kingdom Atomic Energy Authority (NEPTUNE 2057701-TN-03), and the Royal Academy of Engineering (IF-2425-19-AI165).

## Impact Statement

This paper aims to advance machine learning applications in studying the long-term properties of large-scale chaotic systems. There are many potential societal consequences of our work, none which we feel must be specifically highlighted here.

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

## A. Table of Notations

| Notations | Meaning |
|---|---|
| $\mathcal{L}^p(X, \mu)$ | function spaces defined using a natural generalization of the $p$-norm |
| $(X, \mathcal{B}, \mu)$ | measure space |
| $\mu$ | Lebesgue measure |
| $T$ | nonlinear forward map |
| $\theta$ | parameters of neural networks |
| $\phi$ | feature functions in $\mathcal{L}^2$ space |
| $S_i$ | Number of grid points in $i$-th coordinate |
| $N = \prod_{i=1}^{M} S_i$ | Total Number of grid points |
| $C$ | Number of physical quantities |
| $d$ | feature dimensions in $\mathcal{L}^2$ space |
| $k$ | Number of samples from $d-$dimensional unit sphere |
| $m$ | Number of RFF samples |
| $v$ | Random unit vector sampled from $d-$dimensional unit sphere |

## B. Convergence of Random Fourier Features

Let $X \sim N(\mu, \sigma^2)$ be a Gaussian random variable. The characteristic function is defined as:

$$\begin{aligned}
\varphi_X(t) &= \mathbb{E}[e^{itX}] \\
&= \frac{1}{\sqrt{2\pi}\sigma} \int_{-\infty}^{\infty} e^{itx} e^{-\frac{(x-\mu)^2}{2\sigma^2}} \, dx.
\end{aligned} \tag{10}$$

Complete the square in the exponent:

$$\begin{aligned}
itx - \frac{(x-\mu)^2}{2\sigma^2} &= itx - \frac{x^2 - 2\mu x + \mu^2}{2\sigma^2} \\
&= -\frac{1}{2\sigma^2}(x^2 - 2\mu x + \mu^2 - 2it\sigma^2 x) \\
&= -\frac{1}{2\sigma^2}(x^2 - 2(\mu + it\sigma^2)x + \mu^2) \\
&= -\frac{1}{2\sigma^2}((x - (\mu + it\sigma^2))^2 - (\mu + it\sigma^2)^2 + \mu^2)
\end{aligned} \tag{11}$$

Substituting back:

$$\begin{aligned}
\varphi_X(t) &= \frac{1}{\sqrt{2\pi}\sigma} e^{it\mu - \frac{t^2\sigma^2}{2}} \int_{-\infty}^{\infty} e^{-\frac{(x-(\mu+it\sigma^2))^2}{2\sigma^2}} \, dx \\
&= e^{it\mu - \frac{t^2\sigma^2}{2}} \cdot \underbrace{\frac{1}{\sqrt{2\pi}\sigma} \int_{-\infty}^{\infty} e^{-\frac{(x-(\mu+it\sigma^2))^2}{2\sigma^2}} \, dx}_{\text{integral of Gaussian density}}
\end{aligned}$$

The integral equals $\sqrt{2\pi}\sigma$ since it's the integral of a Gaussian density (after appropriate substitution).

Therefore:

$$\varphi_X(t) = e^{it\mu - \frac{t^2\sigma^2}{2}}$$

This completes the proof of the characteristic function for a Gaussian distribution with parameters $\mu$ and $\sigma^2$. In our case, $t$ is replaced by the relative Euclidean norm in the grid domain $\mathcal{M}$ and $\mu$ is zero in random Fourier sampling. Thus, when samples $m$ is sufficiently large, $\theta(i-j)$ in Equation (4) converges to

$$\theta(i-j) = \mathbb{E}[e^{2\pi b(\xi_i - \xi_j)}] = e^{-\pi\sigma^2 \|\xi_i - \xi_j\|_2^2}. \tag{12}$$

## C. Ergodicity and Mixing

Mixing and ergodicity are correlated concepts in the mathematical analysis of dynamical systems (Cornfeld et al., 2012), particularly within the framework of measure theory. Consider a measure-preserving dynamical system $(X, \mathcal{B}, \mu, T)$, where $T : X \to X$ is a transformation that preserves the measure $\mu$. Ergodicity is defined by the property that for any measurable set $A \in \mathcal{B}$, if $T^{-1}(A) = A$ then $\mu(A)$ is either 0 or 1. This implies that time averages equal space averages for integrable functions, formally expressed as

$$\lim_{N \to \infty} \frac{1}{N} \sum_{n=0}^{N-1} f(T^n x) = \int_X f \, d\mu \quad \text{for } \mu\text{-almost every } x \in X.$$

Mixing is a stronger condition where, for any two measurable sets $A, B \in \mathcal{B}$, the measure of their intersection under iteration satisfies

$$\mu(T^{-n}(A) \cap B) \to \mu(A)\mu(B) \quad \text{as } n \to \infty.$$

This signifies that the system asymptotically "forgets" its initial state, leading to statistical independence of $A$ and $B$ over time. The mixing rate quantifies the speed at which this convergence occurs, often characterized by the decay of temporal correlations. Mathematically, if there exists a function $\phi(n)$ such that

$$|\mu(T^{-n}(A) \cap B) - \mu(A)\mu(B)| \leq \phi(n), \tag{13}$$

and $\phi(n)$ decays to zero at a certain rate (e.g., exponentially $\phi(n) \propto \exp(-\lambda n)$), this rate $\phi(n)$ is referred to as the mixing rate. A faster mixing rate implies a more rapid approach to equilibrium, which is crucial for applications ranging from statistical mechanics to the analysis of randomized algorithms. Together, ergodicity ensures the thorough exploration of the state space, while mixing and its associated rate provide a quantitative measure of how efficiently the system achieves statistical uniformity.

## D. Experiment settings.

In this section, we provide the details of numerical experiments and baseline implementations on the high-resolution Kolmogorov flow and turbulent channel flow.

### D.1. Kolmogorov flow

**Governing Equations**   The Kolmogorov flow system is a classic model for studying fluid instabilities and turbulence in two-dimensional incompressible flows (Temam, 2012). It is described by a nonlinear, incompressible Navier-Stokes equation driven by a sinusoidal forcing term.

$$\frac{\partial \mathbf{u}}{\partial t} + (\mathbf{u} \cdot \nabla)\mathbf{u} + \nabla p = \frac{1}{Re}\Delta \mathbf{u} + \mathbf{f} \quad \text{in domain } [0, 2\pi]^2 \times (0, T],$$

$$\nabla \cdot \mathbf{u} = 0 \quad \text{in domain } [0, 2\pi]^2 \times [0, T], \tag{14}$$

where $\mathbf{u}(x, y, t)$ is the velocity field, $p$ is the pressure, and $\mathbf{f} = \begin{pmatrix} 0 \\ \sin(ky) \end{pmatrix}$ represents the external forcing in the $y$-direction. $Re$ is the Reynolds number calculated by $Re = uL/\nu$, where $\nu$ is the kinematic viscosity, $L$ is the characteristic length and $u$ is the speed (Sommerfield, 1908). The equivalent vorticity formulation for $\omega := \frac{\partial \mathbf{u}_y}{\partial x} + \frac{\partial \mathbf{u}_x}{\partial y}$ is

$$\frac{\partial \omega}{\partial t} + \mathbf{u} \cdot \nabla \omega = \frac{1}{Re}\nabla^2 \omega + f \quad \text{in domain } [0, 2\pi]^2 \times (0, T]. \tag{15}$$

We use the spectral method to generate vorticity states with the pseudo-spectral solver in the *jax-cfd* toolbox (Dresdner et al., 2022). The initial conditions follow those initializing methods in (Alieva et al., 2021). We set the maximum speed as $u_{max} = 9.9$ ( $u_x, u_y = 7$ for each direction), $\nu = 1e - 3$, and $k = 4$ on the $256 \times 256$ grid with a temporal resolution of $1.7e - 3$.

**Datasets**   The datasets of vorticity states of the Kolmogorov flow system consist of 150 training trajectories, 40 validation trajectories, and 30 testing trajectories in total. Each trajectory contains 500 frames for 10 seconds with a unique initial state. **KF256** dataset uses the full resolution ($256 \times 256$) of the generated states, and **KF128** dataset downsamples the state resolution by half ($128 \times 128$). Both the ablation studies and the baseline models were trained and evaluated on all of the corresponding sets.

### D.2. Turbulent channel flow

This simulation employs the 3DQ19 lattice model for the lattice Boltzmann method (LBM) (Succi, 2001), which features 3 dimensional and 19 discretized velocity directions. Via Chapman–Enskog analysis, LBM can recover the Navier-Stokes equations and beyond (Succi, 2001). The lattice cell is specified by its position $\mathbf{x}$ at time $t$, and is characterized by a discretized set of speeds $\mathbf{c}_i$ where $i \in \{0, 1, \dots, Q - 1\}$ with $Q = 19$. The evolution equation for the distribution functions can be written as:

$$\mathbf{f}(\mathbf{x} + \mathbf{c}_i\Delta t, t + \Delta t) = \mathbf{f}(\mathbf{x}, t) - \boldsymbol{\Omega}\left[\mathbf{f}(\mathbf{x}, t) - \mathbf{f}^{eq}(\mathbf{x}, t)\right], \tag{16}$$

where $\boldsymbol{\Omega}$ denotes the Bhatnagar-Gross-Krook (BGK) collision kernel (Succi, 2001). The collision kernel relaxes the distribution function towards the local Maxwellian distribution function $f_i^{eq}$:

$$f_i^{eq}(\mathbf{x}, t) = w_i\rho(\mathbf{x}, t)\left[1 + \frac{\mathbf{c}_i \cdot \mathbf{u}(\mathbf{x}, t)}{c_s^2} + \frac{[\mathbf{c}_i \cdot \mathbf{u}(\mathbf{x}, t)]^2}{2c_s^4} - \frac{[\mathbf{u}(\mathbf{x}, t) \cdot \mathbf{u}(\mathbf{x}, t)]}{2c_s^2}\right], \tag{17}$$

where $w_i$ denotes the direction based weights ($w_i = 1/3$ for i=0, 1/18 for the six nearest neighbours and 1/36 for the remaining directions), $\rho(\mathbf{x}, t)$ is the cell macroscopic density, $\mathbf{u}(\mathbf{x}, t)$ is the cell macroscopic velocity. In Equation (16), $\Delta t$ symbolizes the lattice Boltzmann time step, which is set to unity. The LBM kinematic viscosity $\nu$ is defined as

$$\nu = c_s^2\left(\tau - \frac{1}{2}\right)\Delta t, \tag{18}$$

with $c_s$ representing the speed of sound, and $c_s^2$ equating to $1/3$ in Lattice Boltzmann Units (LBU). Now, we apply the Smagorinsky subgrid-scale turbulent model within the LBM framework, the effective viscosity $\nu_{\text{eff}}$ (Smagorinsky, 1963; Hou et al., 1995) is modeled as the sum of the molecular viscosity, $\nu_0$, and the turbulent viscosity, $\nu_t$:

$$\nu_{\text{eff}} = \nu_0 + \nu_t, \qquad \nu_t = C_{\text{smag}} \Delta^2 \left| \bar{\mathbf{S}} \right|, \tag{19}$$

where $\left| \bar{\mathbf{S}} \right|$ is the filtered strain rate tensor, $C_{\text{smag}}$ is the Smagorinsky constant, $\Delta$ represents the filter size. We apply the total viscosity $\nu_t$ into Equation (18) to add the turbulent viscosity. Eventually, macro-scale quantities such as density and momentum are derived from the moments of the distribution function $f_i(\mathbf{x}, t)$, the discrete velocities $\mathbf{c}_i$:

$$\rho(\mathbf{x}, t) = \sum_{i=0}^{Q-1} f_i(\mathbf{x}, t), \tag{20}$$

$$\mathbf{u}(\mathbf{x}, t) = \frac{1}{\rho(\mathbf{x}, t)} \sum_{i=0}^{Q-1} f_i(\mathbf{x}, t)\mathbf{c}_i, \tag{21}$$

**Dataset Contribution**  The datasets are derived from the macroscopic velocity field computed using a lattice Boltzmann numerical solver. A 2D plane ($192 \times 192$) was extracted at the mid-cross-section of the flow direction in a 3D turbulent channel flow. Each pixel in the plane contains velocity vector field information $\mathbf{u}$, including $u_x$, $u_y$, and $u_z$. The datasets comprise 240 training trajectories, 24 validation trajectories, and 24 test trajectories, with each trajectory containing 595 frames. The simulations were executed on 4,096 CPUs over 144 hours, resulting in a total computational cost of approximately 300,000 CPU hours. More details of the dataset are available in Appendix G

### D.3. Baseline implementations

In this subsection, we present the details of the baseline implementations. For all the baselines, the same experimental settings and model configurations are used for both the KF256 and TCF experiments.

- For the U-shaped Fourier Neural Operator (UNO) (Rahman et al., 2022), we adopt the original model configuration described in Rahman et al. (2022) for the Kolmogorov flow with a resolution of $256 \times 256$. The UNO architecture consists of four Fourier blocks, each with 24 frequencies per channel, a width of 64, and a hidden dimension of 96. It includes skip connections and is connected via a channel-wise MLP at the end of each block. The official implementation is utilized from `https://github.com/neuraloperator/neuraloperator/tree/main`.

- For Markov Neural operator (MNO) (Li et al., 2022a), we used the official implementation from `https://github.com/neuraloperator/markov_neural_operator/tree/main`. For the important choice of dissipativity hyperparameters in MNO, we follow their settings in their numerical experiments on Kolmogorov flow with dissipativity regularization coefficient $\alpha = 0.01$, scaling down factor $\lambda = 0.5$, and attractor radius as $156.25 \times S$ where $S$ is the square root of total number of grid points $N$. The MNO model consists of four 2d Fourier layers with 20 frequencies per channel and width = 64.

- Apart from Fourier transform-based baselines, the Multiwavelet-based Operator (MWT) (Gupta et al., 2021) uses wavelet transformations to capture localized, multi-scale dynamics effectively. The MWT model is composed of four 2D multiwavelet blocks, each utilizing 32 Legendre-based wavelets. The network features a hidden dimension of 128, applies ReLU activations between the multiwavelet blocks, and generates outputs through a channel-wise MLP. The official implementation can be found from `https://github.com/gaurav71531/mwt-operator`

- For Factformer (Li et al., 2024), we use the official implementation provided at `https://github.com/BaratiLab/FactFormer/tree/main`. Factformer adopts a scalable architecture combining self-attention mechanisms with low-rank approximations to reduce computational complexities. The factformer consists of four factorized attention blocks. Each attention block has 16[1] heads with dimension 64. The hidden dimension is set to 256 with a two-layered channel-wise MLP as the forward block. To align with other methods, we set the temporal dimension as 1 and follow the *auto-regression* paradigm for training and evaluation as described in (Li et al., 2024).

All models are optimized using the Adam optimizer with the L2 norm between predictions and ground truth, except for MNO. For MNO, the loss function consists of two terms: (1) the first-order Sobolev norm between predictions and ground

---

[1]We increased the number of heads from 8 to 16 to accommodate the higher resolution and align with our implementations.

truth, and (2) a dissipative loss term (refer to (Li et al., 2022a) for details). The learning rate (LR) and scheduler are kept the same as specified in their original implementations. All models are trained for 50 epochs using their default batch size settings.

### D.4. Our transformer implementation details

For the benchmarks, the major hyperparameter configurations of our transformers are listed in Table 5.

*Table 5.* Major hyperparameter configurations of our transformers.

| **Hyperparameters** | **Kolmogorov Flow** | | **Turbulent Channel Flow** |
|---|---|---|---|
| | (KF128) | (KF256) | |
| Number of blocks | 4 | 4 | 4 |
| Attention heads | 8 | 16 | 16 |
| Kernel bandwidth | 8 | 8 | 8 |
| Latent dimension | 256 | 256 | 256 |
| Input function | 2D Conv | 2D Conv | 2D Conv |
| Output function | MLP | MLP | MLP |

The *Latent dimension* determines the number of channels in the latent state representation. The *Number of blocks* specifies the factorized attention blocks incorporated into the encoder of our transformer, influencing its ability to process input data. The *Attention heads* defines the number of heads used in the multi-head attention mechanism, enabling the model to capture diverse patterns effectively. The kernel bandwidth, represented by the sigma parameter of the Gaussian distribution, governs the sampling of the random Fourier spectrum to approximate the kernel. Finally, the operator dimension specifies the latent dimensionality of the bidirectional unitary operator. We use Adam as the optimizer to train the models. The learning rate initiates from $1e-4$ and decays by half for every $10$ of $50$ epochs, following the scales and settings of related chaos works (Li et al., 2022a; 2024). At the end of our model, we employ a three-layer CNN residual block with circular padding to enforce periodic boundary conditions and project back to physical variables. The choice of latent dimension and attention heads follows the setting of the corresponding lowest relative error in the ablation study of (Li et al., 2024).

### D.5. Computational resources for machine learning experiments

Experiments on KF256 and TCF were submitted as resource-restricted jobs to a shared compute cluster. Ablation studies on KF128 were conducted within a group workstation. Computational resources for these machine learning studies used in each chaos system experiment are listed in Table 6.

*Table 6.* Computational resources by experiment.

| **Experiment** | **Hardware** |
|---|---|
| KF128 | 2 GeForce RTX4090 GPUs, 24 GB |
| KF256 | 1 A100 GPU, 40GB |
| TCF | 1 A100 GPU, 40GB |

# E. Evaluation Criteria

In this section, we include further details about the criteria used in Section 4.

## E.1. Short-term accuracy evaluation

**Relative $L^2$ norm.** The stepwise accuracy of the model's predictions for physical states, such as velocity and vorticity, can be evaluated using the relative norm $L^2$, as described in (Li et al., 2022a; 2024). The relative error is defined as

$$\text{Relative } L^2\text{-error}(k) = \frac{1}{N} \sum_{j=1}^{N} \frac{\|\hat{\mathbf{z}}_k^j - \mathbf{z}_k^j\|_2}{\|\mathbf{z}_k^j\|_2}, \tag{22}$$

where $\hat{\mathbf{z}}_k^j$ and $\mathbf{z}_k^j$ are the prediction and ground-truth states at time step $k$ of $j$th test trajectory.

## E.2. Long-term statistics evaluation

**Matching energy spectrum** Evaluation of the matching of the energy spectrum is important to quantitatively assess the model performance (Wan et al., 2023). The energy spectrum is calculated as

$$E(k) = \sum_{|\mathbf{k}|=k} |\hat{\mathbf{u}}(\mathbf{k})|^2 = \sum_{|\mathbf{k}|=k} \left| \sum_{i,j} u(x_{i,j}) \exp\left(-j2\pi\mathbf{k} \cdot x_{i,j}/L\right) \right|^2 \tag{23}$$

where $E(k)$ represents the energy spectrum at wavenumber $k$, and $\hat{u}(\mathbf{k})$ denotes the Fourier transform of the velocity field.
**Mean energy absolute percentage error (ME-APE)**

$$\text{ME-APE} = \frac{1}{N_k} \sum_{k=1}^{N_k} \left| \frac{E_k^{\text{pred}} - E_k^{\text{true}}}{E_k^{\text{true}}} \right| \times 100 \tag{24}$$

where $E_k^{\text{pred}}$ and $E_k^{\text{true}}$ represent the predicted and true energy values, respectively. We evaluate this metric to uniformly evaluate the relative energy spectrum absolute error.

**Mean energy log ratio (ME-LR)**

$$\text{ME-LR} = \sum_K w_k \left| \log\left( \frac{E_{\text{pred}}(k)}{E_{\text{true}}(k)} \right) \right|, \tag{25}$$

where $w_k$ denotes the weight assigned to each wavenumber. This metric measures the logarithmic error between the predicted and true energy values. We further define $w_k = \frac{E_{\text{pred}}(k)}{\sum_k E_{\text{true}}(k)}$ to capture high energy modes as ME-LRw.

**Turbulent Kinetic Energy.** We evaluated the turbulent kinetic energy (TKE) of long-term rollout predictions of baselines and our approach. The TKE represents the mean kinetic energy per unit mass associated with turbulent velocity fluctuations, which quantifies the intensity of the turbulence and the energy distribution across scales. It is defined as:

$$\text{TKE} = \frac{1}{d} \sum_{i=1}^{d} \bar{u}_i'^2,$$

where $\bar{u}_i'^2 = \frac{1}{T} \int (u_i'^2(t) - \bar{u}_i)^2$ and $d$ represent the spatial domain dimensionality. The TKE derived from vorticity field can be approximated from (Cieślak et al., 2019). To evaluate the models, we compute the grid-wise absolute difference between the ground truth TKE and the estimated TKE from the long rollouts of learned models as:

$$\text{Error}_{TKE} = \|\text{TKE}^{\text{true}} - \text{TKE}^{\text{pred}}\|.$$

**Kullback–Leibler Divergence.** We evaluate the statistical distance between the predicted and ground truth distributions of the state field using the Kullback–Leibler (KL) divergence. The KL divergence quantifies the information loss when the predicted distribution $Q$ is used to approximate the true distribution $P$. It is defined as:

$$\text{KL}(P \parallel Q) = \mathbb{E}_P[\log\left(\frac{P}{Q}\right)],$$

where we estimate $P$ and $Q$ via `gaussian_kde` of principal components in samples of the ground truth and predicted fields, respectively. In our setup, the principal components and samples are drawn over the full spatial grid and temporal range of the rollout. A lower KL divergence indicates better agreement in the distributional shape and concentration of vorticity statistics.

**Mixing rate.** The mixing rate of a chaotic system quantifies how quickly its state variables lose the memory of initial conditions and become statistically independent. It measures the decay of correlations and the rate at which distributions approach equilibrium. For a system with state $z$, the autocorrelation function is given as

$$C(t) = \mathbb{E}[T^t(z)z] - \mathbb{E}[T^t(z)]\mathbb{E}[z]$$

The system is said to mix at an exponential rate if $C(t) \propto e^{-\lambda t}$ where $\lambda$ is the mixing rate (see more explanation in Equation (13) at Appendix C). Empirically, we estimated the autocorrelation function up to time-lag $K$ from $N$ true and generated system trajectories such as

$$\hat{C}(t) = \frac{1}{N(T-K-t)}\sum_{i=1}^{N}\sum_{k=0}^{T-K-t}(z_k^i - \bar{z})(z_{t+k}^i - \bar{z}), \quad \text{where } \bar{z} = \frac{1}{NT}\sum_{i=1}^{N}\sum_{t=0}^{T}z_t^i, \tag{26}$$

where $z_t^i$ is the $t$th snapshot in $i$th (true/generated) trajectory. To estimate the mixing rate, we use nonlinear least squares optimization[2] to fit the normalized autocorrelation function $\bar{C}(t) = \hat{C}(t)/\hat{C}(0)$ to an exponential decay model of the form $e^{-\lambda t}$, where $\lambda$ is the fitting parameter. It minimizes the sum of squared errors between the empirical and model values over time lags. This approach allows for a robust estimation of the mixing rate. Finally, we make the absolute difference between the estimated values from ground truth and generated rollouts to form a metric.

---

[2]In practice, we use the `curve_fit` function from the SciPy library (Virtanen et al., 2020), which provides efficient estimation of the parameters by minimizing the residual sum of squares.

# F. Experimental results visualization

In this section, we include more visualized experimental results supplementary to the section 4.

## F.1. Kolmogorov flow

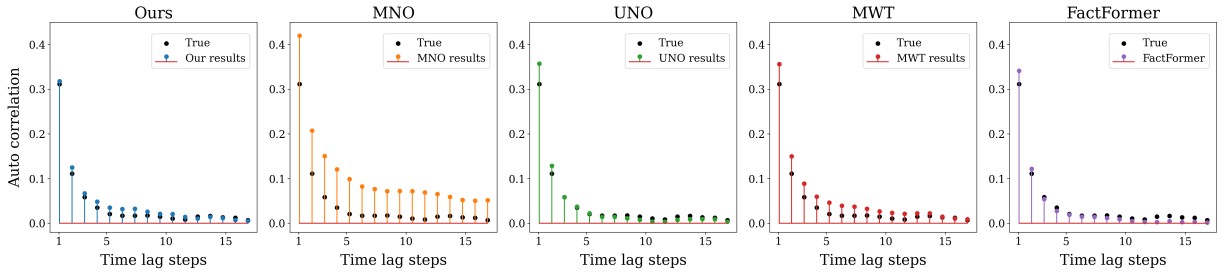

*Figure 4.* Absolute error of short-term KF256 predictions of baselines and ours.

*Figure 5.* Time correlation of our long-term predictions of KF256 and the baselines.

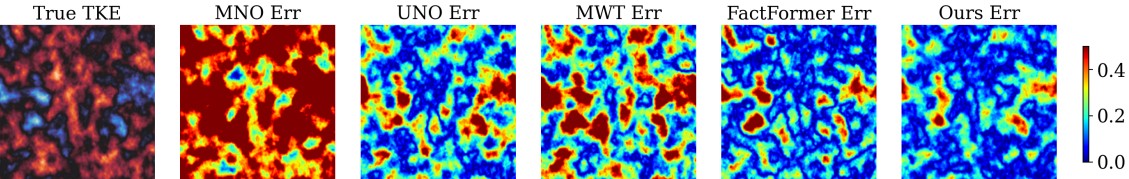

*Figure 6.* TKE error of long-term predictions of KF256. The left-most snapshot indicates the TKE of ground truth trajectories; and the rest snapshots present the absolute percentage error of TKE in 1000-step rollout predictions from baselines and our approach.

## F.2. Turbulent channel flow

For this 3D dataset, the visualization includes short-term prediction states and long-term statistics of the 2D cross-section states in the forward direction.

*Figure 7.* Absolute error of short-term TCF predictions of baselines and ours.

## F.3. Further details on ablation study

In this section, various attention maps of the ablation study section 4.3 on the kernel bandwidth of RFF positional encoding are included in Figure 8.

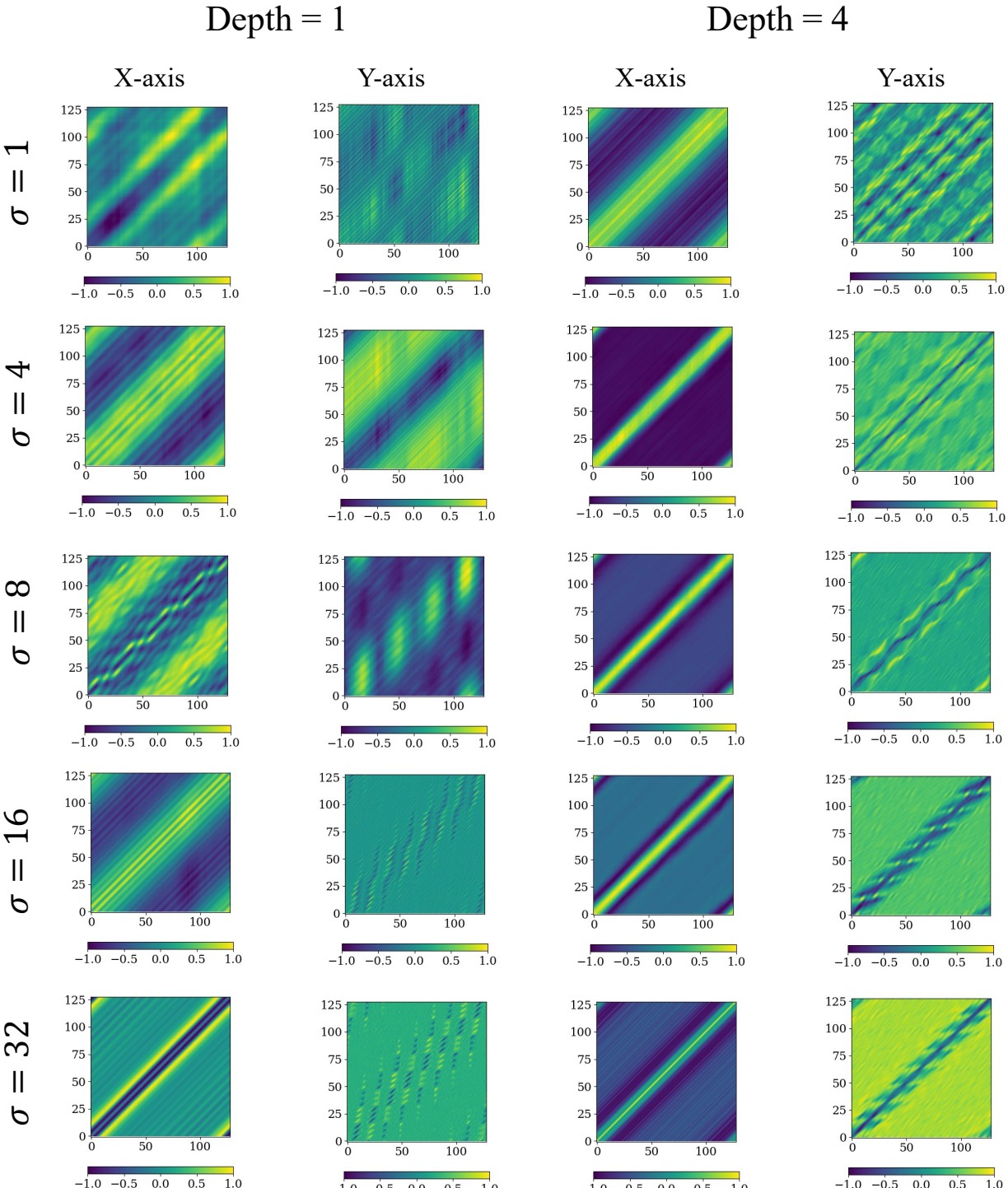

*Figure 8.* Impact of applying random Fourier positional encoding on the attention maps with respect to the kernel bandwidth $\sigma \in \{1, 4, 8, 16, 32\}$. From the attention maps of the first and the last block of our implementation, we observed that a small $\sigma \in \{1, 4\}$ produced overly smoothed attention maps, while the large value $\sigma = 32$ produced noisy attention maps.

## F.4. Computational costs

*Table 7.* Computational costs of our method on scaling grid resolution. We report the model performance of computational runtime, memory usage and floating-point operations per second (FLOPs) per forward pass, across a range of grid resolutions and demonstrated significant time and memory efficiency on increasing resolutions consistently. The results are gathered on the KF datasets given model configurations in Table 5 on a A100 GPU listed in 6 with $7,325,665$ parameters.

| Grid Resolution | Runtime | Memory Usage | FLOPs per Forward Pass |
|---|---|---|---|
| 32×32 | 24 ms | 174 MB | 2.5 GB |
| 64×64 | 27 ms | 537 MB | 10.5 GB |
| 128×128 | 42 ms | 1986 MB | 50.2 GB |
| 256×256 | 58 ms | 7684 MB | 268 GB |

*Table 8.* Computational costs of baselines and ours on TCF predictions. We report the model performance of computational runtime, memory usage and floating-point operations per second (FLOPs) per forward pass. The results are gathered given model configurations in Table 5 on a A100 GPU listed in 6. Attention mechanism is overall computational heavy than neural operator methods, but tensor factorization and axial attention make the computation tractable for large scale chaos states.

| Models | Parameter Count | Runtime | Memory Usage | FLOPs per Forward Pass |
|---|---|---|---|---|
| MNO | 6,467,425 | 31 ms | 377 MB | 3.45 GB |
| UNO | 17,438,305 | 12 ms | 769 MB | 6.88 GB |
| MWT | 5,089,153 | 50 ms | 313 MB | 9.52 GB |
| FactFormer | 6,083,009 | 53 ms | 6889 MB | 239 GB |
| Ours | 7,325,665 | 58 ms | 7684 MB | 268 GB |

## G. Turbulent channel flow as a benchmark for AI4Science

We employ turbulent channel flow as a benchmark to assess the generality of our approach for turbulent flows. Unlike isotropic turbulence, turbulent channel flow is bounded by two parallel planes with no-slip boundary conditions. The Lattice Boltzmann Method (LBM), an alternative to directly solving the Navier-Stokes equations, is inherently parallelization-friendly, making it particularly advantageous for high-performance computing applications. In recent decades, LBM has proven its versatility, enabling applications that range from micro-nano fluidics to macroscopic turbulent flows (Succi, 2001).

In this study, the computational domain for the turbulent channel flow simulation is defined with dimensions $L_x \times L_y \times L_z = 1024 \times 192 \times 192$, where $x$, $y$, and $z$ denote the streamwise, vertical, and spanwise directions, respectively. The friction Reynolds number is set to $Re_\tau = 180$ which is equivalent to $Re = 3250$. Periodic boundary conditions are applied in the streamwise and spanwise directions, while the vertical direction is governed by no-slip boundary conditions (Latt et al., 2008). This configuration distinguishes our dataset from existing studies that primarily rely on 2D turbulent cases, as our dataset is based on fully three-dimensional simulations.

A key motivation for utilizing 3D simulations, even when analyzing 2D cross-sectional results, lies in their ability to capture realistic flow phenomena that are inherently three-dimensional. These include features such as secondary flows, coherent vortical structures, and fully developed turbulence. Such complexities are absent in 2D turbulent channel flow datasets (NVIDIA, 2023), which neglect variations along the third dimension. As a result, 2D datasets fail to adequately represent the intricacies of real-world turbulent flows, limiting their utility in validating models or exploring phenomena where three-dimensional effects are critical. Furthermore, existing high-resolution 3D turbulent channel flow datasets are often derived from direct numerical simulations (DNS), which, while highly accurate, come with significant drawbacks. These datasets can require up to 100 terabytes of storage (JHTDB, 2023), resulting in an unnecessarily high memory cost that makes them impractical for early-stage machine learning research. In comparison, our datasets are designed to be three orders of magnitude smaller, approximately 100 GB, striking a balance between resolution and efficiency. This makes them see for machine-learning-specific tasks, enabling broader accessibility and usability for researchers.

The fully developed 3D turbulent channel flow simulation follows the configuration outlined in reference (Xue et al., 2022). The simulation begins from an initial zero-velocity field, with a square block of size $20 \times 20 \times 100$ grid points positioned at $x = 192$. A volumetric force is applied uniformly across the domain to drive the flow. The simulation is run for 50 domain-through times to establish initial flow characteristics. After this initial phase, the block is removed, and the simulation is continued for an additional 50 domain-through times, allowing the flow to fully develop into a turbulent state. Data sampling begins after this stage, focusing on the 2D cross-section at $x = 512$. Data is collected over a further 100 turnover times, ensuring that the samples represent fully developed turbulence. The simulation timestep is set to $\Delta t = 0.02\,\mathrm{s}$, and it operates in dimensionless units (800 timesteps), as is typical for LBM simulations. The detailed transformation from dimensionless units to physical units can be found in reference (Xue et al., 2024). To ensure robustness and generality, we conducted three independent 3D turbulent channel flow simulations under these conditions, generating a comprehensive dataset for analysis.

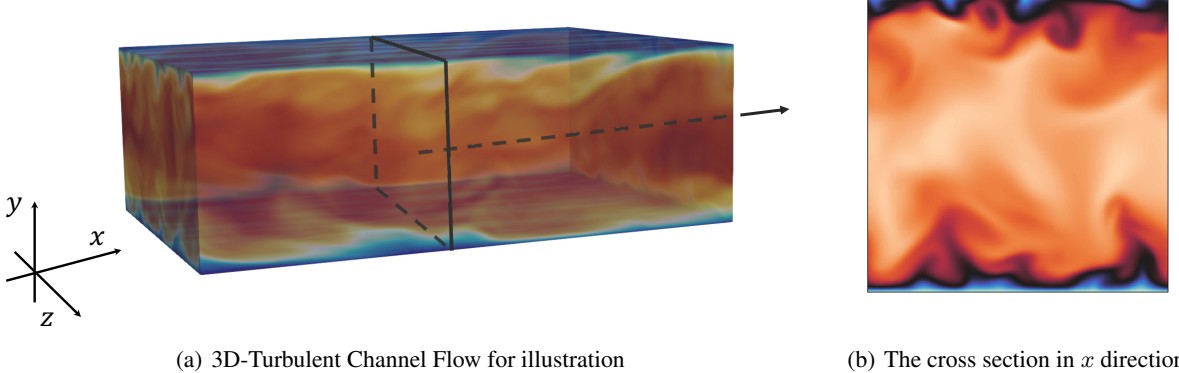

(a) 3D-Turbulent Channel Flow for illustration                    (b) The cross section in $x$ direction

*Figure 9.* Turbulent Channel Flow dataset for illustration

