# OpenReview forum: "Chaos Meets Attention: Transformers for Large-Scale Dynamical Prediction"
_ICML.cc/2025/Conference — ICML 2025 poster_

### Official Review · Reviewer_BWmX · 2025-03-13

**Overall Recommendation:** 3

**Summary:**

The paper addresses the challenging task of accurately forecasting high-dimensional chaotic systems using a transformer-based approach. By leveraging ergodicity and modifying attention mechanisms, the proposed framework effectively handles high-dimensional chaotic dynamics while preserving long-term statistical properties. The model introduces “A3M” attention blocks to capture extreme values and uses a novel loss function inspired by the Von Neumann ergodic theorem to maintain long-term consistency by enforcing unitarity of the dynamics forward map. Experiments on turbulent flows demonstrate superior performance compared to existing methods both in terms of short-term predictions as well as measures of long-term statistics.

**Claims And Evidence:**

Most of the claims are backed by experimental evidence, however, the manuscript lacks some clear information on consistency of the results as well as how fair the comparison is in terms of model expressivity (e.g. by reporting parameter count).

**Essential References Not Discussed:**

The paper misses literature for a specific class of methods to mitigate chaos-induced problems when modeling chaotic DS. Specifically, “Generalized Teacher Forcing” as introduced in Hess et al. (ICML, 2023) does not need to explicitly match distributions of invariant measures to successfully model chaotic dynamics and can be applied to various sequence models, including operator based methods.

**Experimental Designs Or Analyses:**

The authors perform experiments for section 4 for three different random seeds and report the mean in e.g. Table 1 and 2. However, to get a measure of consistency of the results, the manuscript should also report some form of error bars via either standard deviation or standard error of the mean. The same goes for Autocorrelation plots in Fig. 2.

**Methods And Evaluation Criteria:**

The proposed method is only benchmarked on PDE-type data, but lacks evaluation on simpler baselines, such as small chaotic dynamical systems based on ODEs (e.g. Lorenz63 or Rössler system). In these cases, established measures for long-term statistics for *very long* autoregressive rollouts exist, e.g. by comparing Lyapunov spectra. It would be interesting how the model fares in these simpler settings.

**Other Comments Or Suggestions:**

- “These approaches have been applied to classic 1-D examples, such as the Lorenz 63, Lorenz 96, and Kuramoto-Sivashinsky equations.” (p. 1): This may cause confusion as e.g. the Lorenz63 system is a 3D system (three dynamical variables). The manuscript should clearly define what is meant with 1D in this context.

**Typos and similar**:
- “to identifies” (p. 4, l. 177)
- “ergodic measure-preserving transformation (MPT)” (p. 3. ll 161-162): MPT has already been previously defined.
- “The number of sample k” (p. 6, l. 289)
- Kolmogorov is written as “Kolmogrov” on several occasions in the text (e.g. p. 6 l. 321, heading of section 4.1)

**Other Strengths And Weaknesses:**

**Strengths**: The paper clearly motivates each and every compartment of the architecture, combines and advances existing methods in the field and validates everything through comparison methods and ablation studies.

**Questions For Authors:**

1. How exactly is $\mathcal{G}$ implemented? Which form does it take in practice? How expensive is the inversion/complex conjugation of the unitary operator $\mathcal{G}$?
2. how is the Autocorrelation (Fig. 2) computed? Is it pooled over the entire grid?
3. Fig. 3: How long are “1000 steps”? What is the Lyapunov time of the dataset at hand (if such a quantity is easily accessible for this type of PDE data)?
4. Can the architecture be benchmarked on simpler, low-d dynamical systems in forms of ODEs? This should facilitate longer rollouts ($>> 1000$ steps) to check whether the model also captures long-term statistics in these simple cases.
5. Did the authors account for equal expressivity in terms of number of parameters for comparison in e.g. Tables 1 and 2? Can those numbers be reported somewhere?

*I am happy to increase my score if the authors address my concerns*.

**Relation To Broader Scientific Literature:**

The paper presents a novel transformer-based approach specifically designed to improve prediction of high-dimensional and complex chaotic systems with a focus on capturing long-term statistics. The work extends existing approaches by enhancing the traditional transformer architecture with axial mean-max-min (A3M) attention blocks and a unitary operator constraint framework. These innovations enable the model to effectively capture both local extreme values and statistical invariants, thereby improving prediction accuracy over both short-term and long-term horizons. The method is novel in that it introduces a new attention block tailored to chaotic systems prediction as well as combining existing approaches in the operator learning field to enable new state-of-the-art forecasting capabilities.

**Theoretical Claims:**

I did not check the correctness of any proofs or theoretical claims.

---

> ### Author Rebuttal · Authors · 2025-03-31
>
> We genuinely appreciate the reviewer’s time and effort in enhancing our paper.
>
> [Link for results and references](https://anonymous.4open.science/r/ChaosMeetsAttention/README.md)
>
> [Backup link](https://filebin.net/37p4dxup0t320143)
>
> * Concern 1 & Question 5:
>
> `The fair comparison in terms of model expressivity (e.g. by reporting parameter count).`
>
> We appreciate the reviewer for raising concerns similar to those of reviewers vgH9 and ARTA. We report the number of parameters in the provided link. Furthermore, we also compare the computational cost in terms of grid scalability and performer attention, which may also be of interest to the reviewer..
>
> * Concern 2 & Question 4:
>
> `Lacks evaluation on small chaotic dynamical systems based on ODEs.`
>
> Embedding ergodicity into learning methods for chaotic ODEs has been a subject of research and validation in recent literature [1-3], which may be of interest to the reviewer. While the literature did not specifically focus on *large-scale* chaotic dynamical systems, it is one of the primary objectives of our work.
>
> `Establish for very long autoregressive rollouts, e.g. by comparing Lyapunov spectra, on simpler cases.`
>
> As per the reviewer’s request, we implemented MNO [1] and our method on the Lorenz 63 system (L63). We collected 200k rollout steps, along with 300k true velocity data, on the invariant measure of the L63 system. We calculated the Lyapunov exponent for both our predictions and the actual data (1 timestep = 0.05s). Based on this foundation, we present the performance in terms of `Lyapunov exponent` and `Lyapunov time` in the table, and provide additional visualisation results demonstrating the generalizability of our approach on L63 via the provided link.
>
> * Concern 3:
>
> `Clarification of Lorenz 63 dimension`
>
> ‘1D examples’ refers to systems with a one-dimensional spatial domain or systems that are low-dimensional in terms of their structure. We have revised the text to better distinguish between spatial and phase space dimensionality.
>
> `typos`
>
> Thanks for bringing the typos to our attention. We have corrected them in the revised manuscript.
>
> `Reference`
>
> We appreciate the reviewer’s suggestion of the reference, which includes efficient reconstruction approaches through dimension reduction. We have incorporated it into the revised manuscript.
>
> `results consistency & error bar for table 1&2`
>
> The updated tables 1 and 2 are included in the provided link.
> Long-term statistics are robust across multiple runs, where the standard deviations are negligible ($\leq$ 1e-4). Hence, we only report the mean values on the (percentage) advantages. For short-term prediction accuracy, we report the standard deviation in the updated Table 1&2. The results are consistent with the experiment part in the manuscript. We will add the standard deviation to the revised manuscript as suggested.
>
> * Question 1:
>
> `How is $G$ implemented? How expensive is the conjugation of the unitary operator?`
>
> In practice, $G$  is a square matrix in real space. Therefore, the conjugacy relation is equivalent to the transpose $G^T$. The relation between $G$ and $G^T$ preserve the unitarity using our proposed unitary loss in Equation (9) during training. Using the regularized term with Frobenius-norm, the computational complexity is $O(d^3 + 2d^2)$, challenging for large-scale chaos systems. And, we use the Hutchinson trace estimation technique, which effectively accelerates the computation by random projections. This reduces the complexity to $O(kd^2)$ where k is the number of random samples from the $d$-dimensional unit sphere. For further details, please refer to Section 3.3 and Algorithm 1 of the manuscript.
>
> * Question 2:
>
> `Autocorrelation pooled?`
>
> Yes, the autocorrelation is calculated using the equation provided in Appendix E.2, specifically lines 1101-1117. This equation is closely related to the mixing rate. The autocorrelation is then averaged over all spatial points. We appreciate the reviewer’s suggestion to delve deeper into the subject matter.
>
> * Question 3:
>
> `Fig. 3: How long are “1000 steps”?`
>
> Roll out 1000-step counts for 20 seconds for Figure 3. We have provided the dataset details in Appendix G.
>
> `What is the Lyapunov time of the dataset at hand (if such a quantity is easily accessible for this type of PDE data)?`
>
> The Lyapunov time is not directly accessible for the datasets we have. Given the high dimensionality of the data and the time constraints, we will provide a rigorous report of this statistic in the revised manuscript. We would like to know if the reviewer would prefer us to present the Lyapunov times as [mean, maximum] values.
>
> Please let us know if we have addressed the concerns and increased your confidence in our work.

---

> > ### Comment · Reviewer_BWmX · 2025-04-07
> >
> > I thank the authors for the clarifications and additional experiments.
> >
> > **Lyap. Exponent** The results on the Lorenz63 dataset are confusing. In which units are the values reported in the respective table? Can the authors report the maximum Lyapunov exponent in the system's units, i.e. removing discretization? For standard settings ($\rho = 28, \sigma = 10, \beta = 8/3$) we have $\lambda_{max} \approx 0.906$. If one considers the time discretization, one would have $0.05s \cdot 0.906 \approx 0.045s$. Neither of these values are anywhere close to the reported values in table.
> >
> > **Conclusion** Since most of my concerns were addressed and due to the general positive assessment by the other reviewers, I will increase my score. However, I expect the authors to sort the issue with $\lambda_{max}$ out, as in this form the results are not comparable to existing literature.

---

> > > ### Author Response · Authors · 2025-04-08
> > >
> > > We sincerely thank the reviewer BWmX for acknowledging the additional experiments and clarifications, recognizing our improvement, increasing the score and providing insightful feedback on calculating Lyapunov Exponent. These all make the discussion greatly valuable and polish our work to the highest standard.
> > >
> > > We value the insightful comments regarding the computation of the Maximal Lyapunov Exponent (MLE). In response, we carefully re-examined our implementation and identified an error in the normalization during the computation process [Strogatz, 1994]. After correcting this issue, the MLE computed on the ground truth dataset now aligns with values reported in the literature [Viswanath, 1998]. We would like to report the updated results in the table and the visualizations [via the link](https://anonymous.4open.science/r/ChaosMeetsAttention/README.md).
> > >
> > >  Following re-evaluation, Our model achieves an LE of 0.825, which is 19.1% closer to the true value than that of MNO. This improvement is further supported by the evolution of the separation between nearby trajectories in the Lorenz-63 system, initialized at $s_0=[1,1,1]$ with a perturbation vector $\delta_0$ of norm $\delta_0=5e-5$ as shown in the revised figure.
> > >
> > > References
> > >
> > > Strogartz, Steven H. "Nonlinear dynamics and chaos: With applications to physics, biology." Chemistry and Engineering 441 (1994).
> > >
> > > Viswanath, Divakar. Lyapunov exponents from random Fibonacci sequences to the Lorenz equations. Cornell University, 1998.

---

### Official Review · Reviewer_ARTA · 2025-03-14

**Overall Recommendation:** 4

**Summary:**

The paper introduces a transformer-based model for predicting long-term trajectories in high-dimensional chaotic systems. It modifies standard attention mechanisms using Axial Mean-Max-Min (A3M) attention with random Fourier features to capture spatial correlations. It uses a unitary-constrained loss (based on the Von Neumann ergodic theorem) to preserve long-term statistical properties. The approach is scaled efficiently with tensor factorization and shows good performance on chaotic systems.

**Claims And Evidence:**

The paper’s claims are supported by experiments on Kolmogorov flow and turbulent channel flow. Performance improvements, scalability via factorized attention, and ergodicity preservation through a unitary loss are supported by quantitative results. The lack of generalization to non-ergodic systems is acknowledged.

**Essential References Not Discussed:**

One work that might be worth mentioning for context is the Performer (Choromanski et al., 2020), which appears to use random features to achieve linear-time attention.

**Experimental Designs Or Analyses:**

The experimental design seems sound. The paper evaluates its model on two datasets (Kolmogorov flow and turbulent channel flow), using several metrics and three random seeds (though it would be nice to see the standard deviation in addition to the mean). Ablation studies validate the usefulness of key components. More benchmarks would strengthen the paper.

**Methods And Evaluation Criteria:**

Yes. The methods (modified transformer architecture with A3M attention and a unitary loss to preserve ergodicity) make sense for modeling chaotic systems. The evaluation criteria and datasets (Kolmogorov flow and turbulent channel flow) are appropriate for capturing short-term accuracy and long-term statistical behavior.

**Other Comments Or Suggestions:**

The paper is generally well-written, but a few suggestions could improve it further:
- Consider including more details on hyperparameter tuning and reporting variability (e.g., standard deviations).
- Expanding the discussion on applicability to non-ergodic systems would be valuable.
- A clearer explanation of scalability limits and potential computational trade-offs might help.

**Other Strengths And Weaknesses:**

Strengths:
- Combines Transformer architectures with ergodic theory, using A3M attention and a unitary loss to address long-term chaotic dynamics.
- Demonstrates improvements on benchmarks (Kolmogorov flow, turbulent channel flow) and introduces a new turbulent channel flow dataset.
- Includes evaluations and ablation studies that support the key contributions.

Weaknesses:
- The method does not generalize to non-ergodic settings.
- Unitarity is only enforced approximately with soft regularization rather than exactly.
- Performance seems sensitive to the kernel bandwidth.

**Questions For Authors:**

1. Could you elaborate on how to tune the kernel bandwidth in the RFF positional encoding and how its optimal value varries across different datasets?
2. Can you provide more insight into the impact of enforcing unitarity only approximately via soft regularization?
3. Could you clarify the computational trade-offs of your tensor factorization approach compared to other linear-time attention methods (e.g., Performer)?

**Relation To Broader Scientific Literature:**

The paper extends transformer models to chaotic systems by combinding neural operator learning (e.g., Fourier Neural Operators) with ergodic theory (Von Neumann’s theorem). It builds on prior work by addressing scalability in operator-based methods and introduces efficient tensor factorization with the A3M attention mechanism.

**Theoretical Claims:**

I read the derivation using the Von Neumann ergodic theorem, which the paper appears to apply correctly to argue that the operator G should be unitary in L2. However, the paper only enforces this unitarity approximately via a soft loss term (using Hutchinson’s trace), rather than proving exact unitarity.

---

> ### Author Rebuttal · Authors · 2025-03-31
>
> We sincerely thank reviewer ARTA for carefully reviewing our manuscript, providing valuable feedback and reconzing the strengths of our work. We'd like to address your concerns in the initial review and answer you questions as follows:
>
> [Essential Link for results and references to this Rebuttal](https://anonymous.4open.science/r/ChaosMeetsAttention/README.md)
>
> [Backup link](https://filebin.net/37p4dxup0t320143)
>
> * Concern 1 & Question 2 (also mentioned by reviewer vgH9):
>
> `However, the paper only enforces this unitarity approximately via a soft loss term (using Hutchinson’s trace), rather than proving exact unitarity.`
>
> Yes, we use Hutchinson’s trace estimation to approximate unitarity. This approach is motivated by two key reasons:
>
> 1. Directly applying the unitary loss involves eigen decomposition and computing the Frobenius norm, which is computationally infeasible for high-dimensional latent states.
> 2. Hutchinson’s trace estimation is flexible to implement as a randomized projection and is optimized adaptively during training.
>
> Empirically, we demonstrate that the unitarity of the learned operators is well approximated. This is visualized in the **eigenvalues plot** of the learned operators from our ablation models (Base model and +Unitary) in the provided link.
>
>
> * Concern 2 & Question 3:
>
>   `References and computational trade-offs to other linear-time attention methods like Performer.`
>
>   We thank the reviewer for the suggested references and have properly cited them in the final version. We also conducted experiments with Performer attention in our setup (Unitary+Performer) on the TCF benchmark during the rebuttal time. Visualization results are available in the link. The following table compares the **runtime**, **memory consumption**, and **FLOPs per forward pass** of baselines, Performer attention and our method. The results are evaluated on a cuda device with `_CudaDeviceProperties(name='NVIDIA A100-SXM4-40GB', major=8, minor=0, multi_processor_count=108, L2_cache_size=40MB)`. Although attention mechanism is overall computational heavy than neural operator methods, tensor factorization and axial attention manage to make the computation tractable for large scale chaos states. With A3M attention and unitary constraint, our model effectively achieves better performance on the benchmarks. Additionally, we observe that Performer attention trades off memory usage on runtime from the empirical results.
>
>   | Models                 | Parameter count | Runtime    | Memory usage | FLOPs per forward pass |
>   | ---------------------- | --------------- | ---------- | ------------ | ---------------------- |
>   | MNO                    | 6467425         | 31ms       | 377MB        | 3.45GB                 |
>   | UNO                    | 17438305        | 12ms       | 769MB        | 6.88GB                 |
>   | MWT                    | 5089153         | 50ms       | 313MB        | 9.52GB                 |
>   | FactFormer             | 6083009         | 53ms       | 6889MB       | 239GB                  |
>   | Ours                   | 7325665         | 58ms       | 7684MB       | 268GB                  |
>   | **Unitary+Performer** | **7717931**    | **111ms** | **2938MB**  | **271GB**             |
>
> * Concern 3 & Question 1:
>
> `Sensitive to the kernel bandwidth? How to tune it on different datasets.`
>
> We would like to thank the reviewer for raising the concern regarding kernel bandwidth. Through ablation experiments in Section 4.3, we found that selecting the bandwidth within a reasonable interval consistently yields stable results, indicating that tuning the bandwidth is straightforward and efficient, only requiring a coarse cross-validation.
>
> The strategy for tuning the kernel bandwidth depends on the mesh size and viscosity (i.e., Reynolds number) of the system, which help identify a suitable bandwidth interval. If the data characteristics are unfamiliar, we recommend starting with a wider interval and then performing a grid search within that range. We appreciate the reviewer’s suggestion and will include the tuning strategy in the final version of the paper.
>
> * Concern 4:
>
> `Expanding the discussion on applicability to non-ergodic systems would be valuable.`
>
> We thank the reviewer for this suggestion. Extending our methods to non-ergodic systems is a significant open problem. While our current work focuses on ergodic chaotic systems, we agree that exploring non-ergodic systems is an interesting direction for future research.
>
> Finally, we thank reviewer ARTA once again for the time and effort invested in reviewing our paper. We believe the changes made in response to the reviewer’s comments have significantly improved our manuscript. We look forward to your further feedback.

---

### Official Review · Reviewer_FGZz · 2025-03-15

**Overall Recommendation:** 4

**Summary:**

The paper investigates the problem of predicting the evolution of ergodic chaotic systems with transformers. To that end, the paper introduces a set of modifications to the traditional transformer architecture that overcome crucial bottlenecks in terms of scalability. Moreover, the paper introduces a novel regularization term that is grounded in physical perspective and helps the model preserve long-term statistics. The paper evaluates its proposed method on two turbulent fluid dynamics systems (Kolmogorov Flow and Channel Flow) and compares the performance to a multitude of baselines, showing superior performance across all metrics. Finally, the paper releases its data as a novel chaotic system benchmark.

**Claims And Evidence:**

The paper is exceptionally well written. All claims are supported by clear and convincing evidence.

**Essential References Not Discussed:**

While not directly related in terms of the type of chaotic system evaluated, the paper would benefit from citing Gilping (2021, 2023), who evaluates transformers on more than 130 chaotic dynamical systems, and Zhang et al. (2025), who evaluate transformers on elementary cellular automata.


**References**

Chaos as an interpretable benchmark for forecasting and data-driven modelling.
William Gilpin.
NeurIPS 2021.

Model scale versus domain knowledge in statistical forecasting of chaotic systems.
William Gilpin.
arXiv:2303.08011

Intelligence at the Edge of Chaos.
Shiyang Zhang, Aakash Patel, Syed Rizvi, Nianchen Liu, Sizhuang He, Amin Karbasi, Emanuele Zappala, David van Dijk.
ICLR 2025

**Experimental Designs Or Analyses:**

The paper presents the first evaluation of transformer-based methods on (ergodic) chaotic systems and justifies this choice with strong empirical results. The paper proposes a set of modifications to the standard transformer architecture: (i) axial mean-max-min (A3M0 attention, (ii) 2D positional encodings based Random Fourier Features (RFF), and (iii) a loss regularization term based on a unitary constraint. Most of these design choices are either empirically validated (for i) or theoretically motivated (for iii).

However, it would be nice to also show an ablation for the distance-based Gaussian kernel, i.e., A3M without RFF (Table 3 only ablates the full A3M). Moreover, the paper showing the running times in Table 3 would further strengthen the paper's claims.

**Methods And Evaluation Criteria:**

The paper evaluates its proposed method on two turbulent fluid dynamics, Kolmogorov Flow and Channel Flow, both commonly used to learn states from chaotic systems. Moreover, the paper considers a variety of competitive baselines, both operator- and attention-based. As a result, the experimental evaluation is convincing and shows strong results.

**Other Comments Or Suggestions:**

List of typos and mistakes:
* L177 (left): "an attention mechanism to identif**ies**" -- replace with "identify"
* L178 (left): "based on **random Fourier using** Random Fourier features" — replace duplicates
* L445 (left) "none **of** which"

**Other Strengths And Weaknesses:**

The benchmark dataset will perhaps be the paper's biggest contribution in the long run and should, therefore, be featured more prominently (i.e., in the main paper).

**Questions For Authors:**

None

**Relation To Broader Scientific Literature:**

The paper adequately discussed the related work.

**Theoretical Claims:**

I am unfamiliar with the mathematical machinery required to evaluate the correctness of the theoretical claims.

---

> ### Author Rebuttal · Authors · 2025-03-31
>
> We sincerely appreciate reviewer FGZz for taking the time to thoroughly review our manuscript, offering valuable feedback, and acknowledging the strengths of our work, particularly in the areas of efficient transformers for large-scale chaos systems, physics-inspired regularisation terms, and the contribution of our dataset to the community. Below, we respond to each of the reviewer's comments and suggestions.
>
> - **Ablation on distance-based Gaussian kernel**
>
> We thank the reviewer's valuable suggestion. It should be clarified that we actually included an ablation study of Gaussian kernels in Section 4.3 (Table 4) and Appendix D (Figure 4). In particular, we observed that extremely large sigma values like 32 lead to a rapid decay of spatial correlations, which is effectively equivalent to the case without a kernel. We will clarify this in the revised version.
>
> Regarding computational efficiency, we evaluated the **runtime** per batch for the ablation models presented in Table 3. The results are evaluated on a cuda device with `_CudaDeviceProperties(name='NVIDIA GeForce RTX 4090', major=8, minor=9, total_memory=24GB, multi_processor_count=128, L2_cache_size=72MB)` with a batch size of 3 over 100 runs.
>
>   | Method           | Runtime (per batch, size=3) |
>   | ---------------- | --------------------------------|
>   | Base               | 0.0261 ± 0.00016                 |
>   | + A3M Att.      | 0.0244 ± 0.0008                   |
>   | + Unitary Op.  | 0.0995 ± 0.0077                   |
>
> - **Suggested References**
>
> We appreciate the reviewer’s recommendations. After careful consideration, we’ve integrated the relevant references into the introduction section of our revised manuscript.
>
> - **Benchmark Dataset**
>
> We appreciate the reviewer’s acknowledgement of the significance of our proposed benchmark dataset. We will highlight this contribution even more in the final version of our manuscript.
>
> - **Typos**
>
> We appreciate the reviewer's meticulous review and for bringing the typos to our attention. All the identified typos have been meticulously corrected in the revised manuscript.
>
> We sincerely thank the reviewer once again for their insightful comments and suggestions. We believe that the revisions made in response have significantly improved the quality of our manuscript. We eagerly anticipate any further feedback. In the meantime, we have prepared new visualisations and the code related to trying performer attention via the link [here](https://anonymous.4open.science/r/ChaosMeetsAttention/README.md), which may be of interest to the reviewer.

---

> > ### Comment · Reviewer_FGZz · 2025-04-02
> >
> > **We observed that extremely large sigma values like 32 lead to a rapid decay of spatial correlations, which is effectively equivalent to the case without a kernel. We will clarify this in the revised version.**
> >
> > Indeed, I hadn't considered that. Thanks for clarifying!
> >
> > I still think that it would be interesting to see the case without the distance-based Gaussian kernel, but I guess Table 4 is already a good start.
> >
> >
> > **Running times**
> >
> > Thanks, that's great to see!

---

> > > ### Author Response · Authors · 2025-04-05
> > >
> > > Many thanks for recognizing our reply and your further guidance on the ablation study to us. We dropped the distance-based Gaussian kernel and re-trained the model using the same settings as in Table 4 of the ablation study. The updated results are summarized below:
> > >
> > > | τ = 5 | τ = 25 | ME-APE | ME-LRw | Δλ |
> > > | ------ | ------- | ------ | ------ | ---- |
> > > | 0.93   | 1.31    | 0.19   | 0.21   | 0.11|
> > >
> > > We are diligently working to refine the manuscript and address your comments thoroughly. Please let us know if you have any further feedback or updates.

---

### Official Review · Reviewer_vgH9 · 2025-03-23

**Overall Recommendation:** 3

**Summary:**

The paper introduces a transformer-based framework for predicting large-scale chaotic systems. The authors tackle a key challenge in dynamical system forecasting -- the amplification of prediction errors due to positive Lyapunov exponents -- by using ergodicity. Their approach includes:

- A modified attention mechanism (A3M Attention) that captures statistical moments and extreme values in chaotic systems.
- A loss function inspired by the von Neumann mean ergodic theorem, aimed at enhancing long-term statistical stability.
- A large-scale dataset featuring 140k snapshots of turbulent channel flow to benchmark prediction accuracy.

**Claims And Evidence:**

The paper presents compelling claims regarding the effectiveness of its transformer model in maintaining long-term statistical properties while enhancing short-term prediction accuracy. While the claims are well-supported by empirical results and the von Neumann ergodic theorem is leveraged to justify the approach, it remains unclear whether the proposed loss function strictly enforces ergodicity in practice; thus, additional theoretical validation, such as a formal proof or convergence analysis, would further strengthen the argument.

**Essential References Not Discussed:**

I also recommend considering the following papers on learning chaotic dynamics with RNNs and combining deep learning techniques with Koopman operator theory:
1. https://proceedings.mlr.press/v202/hess23a/hess23a.pdf
2. https://arxiv.org/pdf/2410.23467

Also, the paper does not sufficiently discuss PINNs and other hybrid methods that integrate physics-based constraints into ML models. Works such as:
1. Karniadakis et al., "Physics-Informed Neural Networks for PDEs" (Journal of Computational Physics, 2021)
2. Raissi et al., "Physics-Informed Machine Learning for Dynamical Systems" (PNAS, 2019).

**Experimental Designs Or Analyses:**

The experimental design is robust, utilizing diverse datasets that represent various chaotic systems and comparing against multiple state-of-the-art baselines. However, a more detailed computational complexity analysis comparing memory consumption and training time across baselines would strengthen the overall evaluation of the model's efficiency.

**Methods And Evaluation Criteria:**

While the evaluation criteria used in the paper (relative $L^2$ norm for short-term accuracy, ME-APE, ME-LRw, and $\nabla \lambda$ for long-term statistical consistency) are reasonable, additional and more rigorous evaluation metrics could provide a more comprehensive assessment of the model's performance. Could the authors consider integrating some of these additional metrics to provide a more holistic evaluation of their method? For instance:

- Kullback-Leibler (KL) Divergence or Wasserstein Distance, or/and the Hellinger distance: Please see Sect. 4.2  of this paper for more details https://proceedings.mlr.press/v202/hess23a/hess23a.pdf

- Autocorrelation Decay Rate: Measuring how the model preserves the autocorrelation structure over time can indicate its ability to maintain memory of system dynamics.

- Computational Cost-to-Accuracy Ratio: A more explicit comparison of runtime, memory usage, and FLOPs per training iteration would provide a clearer picture of the model’s scalability.

And maybe this one:

- (Max) Lyapunov Exponent Deviation: Evaluating how well the predicted trajectories preserve the Lyapunov exponents (or max LE) of the true system would provide a more direct measure of long-term dynamical consistency.

**Other Comments Or Suggestions:**

It would be helpful to:
- Elaborate on the strategies used for hyperparameter tuning of random Fourier features.
- Provide a comprehensive comparison of runtime and memory usage with baseline methods.

I am happy to increase my score if the authors can address my main concerns.

**Other Strengths And Weaknesses:**

**Strengths**

Originality: The introduction of A3M Attention and ergodicity-based loss function presents a novel approach to chaotic system modeling.

Significance: The paper tackles an important problem in ML-driven dynamical system forecasting with well-motivated solutions.

**Weaknesses**
- The paper mainly focuses on comparing transformer-based and operator-based models but does not discuss how physics-informed neural networks (PINNs) or hybrid methods (e.g., physics-constrained transformers) might perform on chaotic systems.
- Choosing the kernel bandwidth ($\sigma$) in RFF positional encoding may significantly affect performance. It would be helpful to have a more in-depth discussion on strategies for tuning hyperparameters.
- Even though the paper mentions improved efficiency, a deeper look into complexity - like memory usage and training time comparisons -
 would provide more clarity.

**Questions For Authors:**

1. How does the von Neumann ergodic loss stack up against traditional loss functions when it comes to stability and long-term statistical preservation? Also, how does the ergodicity-based loss function compare to traditional distribution-matching methods in terms of computational efficiency?
2. How does the A3M attention mechanism perform compared to other modifications of attention (e.g., Linformer, Performer, or Reformer) in chaotic dynamics?
3. Regarding the computational efficiency of your approach, how does training time scale with increasing grid resolution?
4. Given that $\sigma$ in RFF positional encoding may significantly affect performance, how sensitive is the model to variations in $\sigma$ across different datasets? How sensitive is the model to hyperparameter tuning, particularly for the kernel bandwidth σ\sigma in the positional encoding?
5. What if we swapped out the A3M pooling approach for a different aggregation method? Would using just max-pooling or mean-pooling give us competitive results?
6. Can you provide scalability benchmarks comparing the runtime and memory usage of your method against baselines?

**Relation To Broader Scientific Literature:**

The paper is well-situated within the literature on chaotic system prediction, referencing key works on operator-based learning, transformers for PDEs, and ergodicity-based learning approaches. It builds upon advancements in Fourier Neural Operators (FNOs) and Markov Neural Operators (MNOs) while introducing a novel transformer-based framework specifically designed to enhance prediction accuracy while preserving ergodicity, a crucial property highlighted by Eckmann & Ruelle (1985) and Young (2002). Moreover, the introduction of a new chaotic system benchmark dataset enriches the field by providing a standardized resource for evaluating machine learning methods in chaotic dynamics. These contributions collectively advance the intersection of machine learning and dynamical systems theory, offering both methodological improvements and practical benchmarks for future research.

**Theoretical Claims:**

The paper provides a well-motivated theoretical foundation using the von Neumann mean ergodic theorem, but key aspects require further clarification. The claim that the operator $G$ is unitary and preserves norms in $L^2$ space is critical, and verifying whether these properties hold under the proposed modifications is essential for the method's validity. The regularization term in the loss function aims to enforce unitarity, but assessing whether it sufficiently guarantees ergodicity in long-term predictions is crucial. Also, while the authors argue that their approach captures invariant statistical behaviors, a more rigorous justification -- particularly for high-dimensional chaotic systems --
 would strengthen confidence in their claims regarding stability and accuracy.

---

> ### Author Rebuttal · Authors · 2025-04-01
>
> We greatly appreciate the reviewer's constructive feedback and recognition of our work. We address the concerns and questions as follows:
>
> [Link to visualizations and references](https://anonymous.4open.science/r/ChaosMeetsAttention/README.md)
>
> [Backup](https://filebin.net/37p4dxup0t320143)
>
> * Concern 1 & part of Question 1:
>
>     `Whether the proposed loss function strictly enforces ergodicity in practice, any proof, and preserves in long-term predictions.`
>
> We thank the reviewer for this insightful question. The ergodicity guarantee through the unitarity constraint is supported by Von Neumann’s theorem (cited in Lines 156-160). Imposing hard unitarity constraints requires complex parameterisation and intensive matrix decomposition during training, which are computationally intensive. Instead, we use Hutchison trace estimation to efficiently approximate the constraint through random projections, enabling its application in large-scale chaotic dynamics. Empirical results show that the learned operators maintain unitary, as visualized in the eigenvalues plot (see link). To verify long-term stability, we use the metrics suggested by the reviewer (next section).
>
> * Concern 2:
>
>     `KL Divergence and other metrics`
>
> We thank the reviewer for the reference. We evaluated the **KL divergence (KLD)** of model predictions on both datasets (KF256 and TCF). Updated Tables 1 and 2 are presented in *Section B: Updates to Reviewer BWmX* (see link). Our method achieves the lowest KLD, demonstrating its effectiveness in preserving long-term statistics and capturing the underlying invariant distribution.
>
> `Autocorrelation Decay Rate & (Max) Lyapunov Exponent Deviation`
>
> We appreciate the suggestion of these metrics. The autocorrelation decay rate, which relates to the mixing rate, is presented in Appendix E, with visualizations in Figures 3 and 6. Estimating the Lyapunov exponent for high dimensional PDE system was challenging. Instead, we provide the estimated Lyapunov exponent deviation for a lower-dimensional case using our method (see link).
>
> * Concern 3 & Questions 2, 3, 6:
>
>   `Computational cost (runtime, memory, FLOPs) and scaling with grid size`
>
> We collected relevant results and organized them into two informative tables (see link). Analytical insights on attention mechanisms are shared in *Response to Reviewer ARTA: Concern 2*. Our method demonstrates linear increase of computational cost in grid size, indicating the strong scalability and computational efficiency of A3M attention.
>
> * Concern 4:
>
>   `Relevance to PINN and physics hybrid transformer`
>
> We have added a discussion on this aspect in the manuscript. PINN methods require explicit knowledge of differential equations, which are assumed to be unknown in our setting. In contrast, our approach only assumes ergodicity, without relying on such prior knowledge, and focuses on embedding physical properties within transformer architectures. Our method provides a distinct perspective from PINN in incorporating physical knowledge.
>
> * Concern 5 & Question 4:
>
>  `Tuning strategy for kernel bandwidth`
>
> Due to space limits, we kindly refer the reviewer to *Response to Reviewer ARTA: Concern 3*. The ablation study on kernel bandwidth (Table 4, Appendix C) shows that selecting a reasonable interval yields stable results.
>
> * Question 1:
>
> `Ergodic loss vs traditional methods`
>
> We appreciate the reviewer’s insightful questions regarding the ergodic loss.
>
> (1) The ergodic loss is designed to enforce unitarity in the forward operator, thereby maintaining energy throughout forecasting. This is crucial for stability and long-term statistical consistency in chaotic systems. Traditional loss functions like MSE focus on short-term accuracy, which can lead to model drift and instability in long-rollout scenarios.
>
> (2) Compared to traditional distribution-matching methods [5,6], our ergodic loss demonstrates superior computational efficiency and scalability. Distribution-matching methods like MMD [1,3] involve kernel operations that scale poorly with sample size, typically requiring large batches for high-resolution data and leading to significant memory consumption. Moreover, MMD with kernels (e.g., rational quadratic, RBF) requires careful bandwidth tuning, as results are sensitive to bandwidth choices [2].
>
> Our ergodic loss enforce unitary dynamics using matrix operations without such hyperparameter tuning. Using stochastic Hutchinson trace estimation, we reduce the computational complexity to $\mathcal{O}(kd^2)$, where $d$ is the latent dimension and $k$ is the number of random draws from the unit sphere [4]. Typically, $d \ll D$ (the system dimension), and the convergence rate scales as $\mathcal{O}(1/\sqrt{k})$. In practice, we found $k \approx 1000$ sufficient for $d = 256$, balancing accuracy and computational cost.
>
> * Question 5:
>
>   `Alternative pooling methods`
>
> We implemented the mean-pooling method for comparison with our full A3M method. Results are reported in the link.

---

### Decision · Program_Chairs · 2025-05-01

**Decision:**

Accept (poster)

**Comment:**

This submission received four expert reviews, with all reviewers acknowledging the authors' rebuttal. The panel agreed that this is a high-quality paper tackling an important ML problem with novel ideas, excellent presentation, very systematic methodology and exemplary testing/validation of every component of the proposed methodology. The reviewers also praised the originality of the proposed modification to the transformer architecture, and the used benchmark dataset which needs to receive more attention in the main part of the paper as it is likely to attract more attention in the future.

The author-reviewer discussion led to a very constructive exchange with the authors presenting high-quality additional material addressing many of the reviewers initial concerns (or requests for clarifications). I strongly encourage the authors to revise the paper along these lines, i.e., to include the new insights/results that are present in the author-reviewer discussion. This will substantial benefit the paper and strengthen its contribution and visibility in the ML community.